# Towards Adaptive Symmetry Breaking in Vector Neuron Networks

## Abstract

Vector Neuron Networks (VNNs) have been widely adopted in various 3D tasks due to their data efficiency and strong generalization capabilities rooted in equivariance. However, the rigid equivariance constraints of VNNs limit their ability to handle the prevalent problem of symmetry breaking in the 3D world, where models may need to produce outputs with reduced symmetry from inputs with high symmetry. In this paper, we propose an adaptive equivariance paradigm within the Vector Neuron (VN) framework, comprising three key designs: (1) Architecturally, we introduce a residual architecture that transforms the rigid equivariance constraints of VNNs into soft priors, preserving their symmetry-based inductive bias while enabling symmetry breaking. (2) Methodologically, we derive an implicit equivariance regularization method that allows VNNs to dynamically adjust their equivariance constraints according to the symmetry level of input data. (3) Structurally, we design a lightweight and interpretable module that allows VNNs to regulate equivariance in a simpler and more transparent manner. Experiments on 26 categories with varying input symmetries demonstrate that our approach achieves adaptive equivariance, improving the average performance of VNNs on pose estimation tasks by a factor of 5, and by a factor of 33 on highly symmetric inputs. Code of our method is available at anonymous repository https://github.com/anon-mity/Ada-VNNs.

## 1 Introduction

$\mathbf{S}$YMMETRY refers to the property of an object or system that remains unchanged or invariant under certain transformations Weyl (2015); Je et al. (2024). Incorporating symmetry as an inductive bias in machine learning has emerged as a powerful paradigm, leading to significant conceptual and practical advances Bronstein et al. (2021). In the 3D domain, VNNs Deng et al. (2021); Katzir et al. (2022); Assaad et al. (2023) have become a highly promising class of SO(3)-equivariant models Thomas et al. (2018); Esteves et al. (2020); Chen et al. (2021); Zhu et al. (2023); Shen et al. (2024); Liu et al. (2025), particularly for symmetric geometric and point cloud data. Traditional neural networks require numerous rotated examples to learn such symmetries, whereas VNNs embed them by network structure design, allowing the model to focus directly on learning the intrinsic task mapping.

Equivariant networks are built upon symmetry assumptions and therefore cannot model symmetry breaking problem Xie & Smidt (2024); Smidt et al. (2021). Specifically, when presented with inputs of higher symmetry, they are incapable of producing outputs with lower symmetry Lawrence et al. (2025). However, many real-world systems and tasks exhibit spontaneous symmetry breaking. A typical example arises in pose estimation task involving rotationally symmetric objects. A Coca-Cola can is almost perfectly rotationally symmetric around its vertical axis. Yet opening the can requires identifying the precise orientation of the pull tab, which constitutes a functionally critical asymmetry. In such cases, the model must make a biased decision from a highly symmetric input.

Unfortunately, due to the rigid structural constraints imposed by predefined operations, VNNs struggle to adjust their degree of equivariance, making it difficult to handle inputs with high symmetry. This raises a fundamental question: *Is there an adaptive* SO(3)-*equivariant model that relaxes equivariance constraints based on the symmetry level of the input data, thereby achieving flexibly symmetry breaking*?

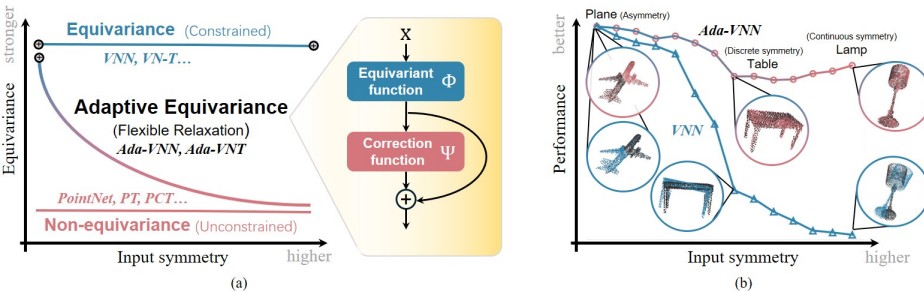

Figure 1: (a) Our adaptive paradigm combines the strengths of both equivariant and non-equivariant approaches. It flexibly relaxes the model's equivariance constraints according to the input's symmetry. (b) Comparison of pose estimation performance between Ada-VNN and VNN on inputs with different symmetries. Ada-VNN effectively handles high symmetry inputs.

In this work, we propose Adaptive Vector Neuron Networks (Ada-VNNs), the first flexible relaxation method designed for VNNs (see Fig.1). It consists of three key components. Architecturally, we introduce a residual connection architecture within the VN framework, defined as follows: $f(x) = \Phi(x) + \Psi(\Phi(x))$. Here, the strictly equivariant function $\Phi$, based on the VN framework, is connected via a residual path to a correction function $\Psi$ with learnable equivariance. By feeding equivariant features into $\Psi$ and combining them through residual addition, our method retains symmetry-based inductive bias while allowing correction function to adaptively capture complementary geometric information beyond the symmetry constraints. This architecture is inspired by ResNet He et al. (2016) and Residual Path Priors (RPP) Finzi et al. (2021), and is effective in transforming the hard equivariance constraints of VNNs into soft priors. Thus, it lays the foundation for breaking equivariance constraints in VNNs.

Rigidly discarding part of the equivariance constraints (e.g., Huang et al. (2023); Fu et al. (2024)) was suboptimal, as inputs with varying degrees of symmetry require different levels of constraint relaxation. Methodologically, to enable flexible symmetry breaking, we derive an implicit regularization method from the objective function of the $\mathrm{SO}(3)$-equivariant task. This method adaptively adjusts the equivariance constraints of the overall function $f$ during training, based on the symmetry level in the data, without requiring any additional equivariance loss terms. Specifically, when the input exhibits low symmetry, the implicit regularization is stronger, encouraging the correction function $\Psi$ to develop equivariance, thereby maintaining strong equivariance in the overall function $f$. In contrast, when the input exhibits high symmetry, the regularization weakens, and the $\Psi$ is no longer sufficiently regularized during training, allowing the equivariance of the $f$ to relax. This leads to a self-organizing adaptive behavior, where the model's equivariance is spontaneously adjusted according to the symmetry characteristics of the training data.

Structurally, we propose a lightweight and interpretable Decoupled Vector MLP (DV-MLP) as the correction function $\Psi$. While maintaining the parameterization lightweight, it explicitly reflects the degree of equivariance relaxation through the distance between the weight matrices of linear layers. This enables a more controllable adjustment of the model's equivariance during training.

Experimental results show that in $\mathrm{SO}(3)$-equivariant tasks, the equivariance constraint of Ada-VNNs can adaptively relax based on the symmetry degree of the input data, and effectively addresses the symmetry breaking problem. Moreover, our method is lightweight and efficient—on pose estimation task involving 40 ModelNet categories and 26 ShapeNet categories, it incurs only about 0.2% additional GPU memory overhead while yielding remarkable average performance gains of approximately 508% and 70%, respectively. To sum up, our contribution are:

1. **From Hard Constraints to Soft Priors**. Through the residual connection within the VN framework, we transform the hard structural constraints of VNNs into soft priors, endowing them with the ability to break equivariance constraints.

2. **From Rigid Relaxation to Flexible Relaxation**. We derive an implicit equivariance regularization method that provides a practical mechanism for adaptively adjusting the model's equivariance constraints in response to varying levels of input symmetry.

3. **Lightweight and Interpretable**. We design a lightweight DV-MLP that allows Ada-VNNs to adjust equivariance constraints more efficiently, while providing transparent and interpretable control throughout training.

## 2 RELATED WORKS

### 2.1 SO(3)-EQUIVARIANT NEURAL NETWORKS

Embedding the symmetry of 3D data is a key principle in point cloud network design, as it significantly impacts performance across various tasks. Among these, SO(3)-equivariance has received considerable attention. Early models such as SphericalCNN Cohen et al. extended planar equivariance Cohen & Welling (2016) to SO(3), while others Chen et al. (2021); Worrall & Brostow (2018) approximated it via finite subgroups (e.g., icosahedral, octahedral) for tractable convolutions at reduced expressiveness. Full SO(3)-equivariance was later achieved using irreducible representations and high-order tensors Thomas et al. (2018); Fuchs et al. (2020), or Frame Averaging Atzmon et al. (2022) which faces continuity constraints Dym et al. (2024). Alternatively, VNN Deng et al. (2021) replaced scalar with vector neurons, and REQNN Shen et al. (2024) adopted quaternions. However, relying on rigid, predefined operators limits flexibility, leaving these models unable to handle the symmetry breaking often seen in real-world 3D tasks.

### 2.2 APPROXIMATELY AND RELAXED EQUIVARIANT NETWORKS

To address this, recent works explore piecewise Atzmon et al. (2024) or relaxed equivariance. For instance, Kaba & Ravanbakhsh (2023) defines relaxed equivariance and argues that such models handle symmetry breaking effectively. van der Ouderaa et al. (2022) formalizes approximate symmetries by allowing small equivariance errors. EQ-REG Bai et al. (2025) applies the approximate equivariant network to image restoration. In reinforcement learning, RPP Finzi et al. (2021) introduces residual path priors that combine equivariant and non-equivariant branches, relaxing strict equivariance into a soft prior. Duet Suau et al. (2023) and CARE Gupta et al. (2024) learn approximately equivariant latent representations for self-supervised learning. Elsayed et al. (2020) showed that relaxing spatial weight sharing in CNNs improves classification accuracy. Building on these, Wang et al. (2022) relaxed weight sharing constraints in group and steerable CNNs to approximate SO(2) and E(2) equivariance. Inspired by these studies, we implement adaptive relaxation based on the input's symmetry level within the VN framework.

### 2.3 VECTOR NEURON NETWORKS

VNN Deng et al. (2021) replaced traditional scalar features with vector neurons and designed core components—e.g., linear layers, nonlinearities, and normalization—to be equivariant under SO(3). Owing to its simplicity and practical effectiveness, VNN has since been extended to SE(3) (SPD-VN Katzir et al. (2022)), SIM(3) (EFEM Lei et al. (2023a)), graphs (E-GraphONet Chen et al. (2022)), and attention (VN-T Assaad et al. (2023)), with applications spanning classification Zhu et al. (2022), segmentation Lei et al. (2023b), registration Yao et al. (2025), pose estimation Yang et al. (2024c), and robotics Yang et al. (2024b). OAVNN Balachandar et al. (2022) was the first to address symmetry breaking in VNNs, resolving reflection ambiguity via planar symmetry detection, but its scope is limited to mirror symmetry and point cloud segmentation. In contrast, we propose the first adaptive framework that effectively handles symmetry breaking under SO(3).

## 3 BACKGROUND

In this section, we clarify the symmetry breaking problem from the perspective of group theory, starting with the definition of symmetry:

**Definition 1.** (Symmetry) An input $x \in X$ is called symmetric if its stabilizer subgroup: $\text{Stab}_G(x) = \{g \in G \mid \rho(g)x = x\}$ is non-trivial, here $\rho(g)$ denotes the action of $g$ on $x$. That is, a symmetric input is fixed by multiple group elements, including at least one non-identity element.

Next, we introduce the following observation first proposed in Smidt et al. (2021).

**Lemma 2.** *(Equivariance constraint). Let $X$ and $Y$ be spaces equipped with a group action of $G$. We can choose an equivariant $f : X \to Y$ such that $f(x) = y$ only if $\mathrm{Stab}_G(y) \supseteq \mathrm{Stab}_G(x)$.*

This implies that the symmetry of the output of an equivariant function cannot be lower than that of the input. The proof of this lemma can be found in Xie & Smidt (2024) Section E.1. Following Xie & Smidt (2024), we can provide a definition of symmetry breaking at the sample level.

**Definition 3.** *(Symmetry breaking). Let $G$ be a group. A sample with the input $x$ and output $y$ is symmetry breaking with respect to $G$ if $\mathrm{Stab}_G(y) \not\supseteq \mathrm{Stab}_G(x)$.*

Thus, $G$-equivariant models are inherently incapable of handling symmetry breaking samples with respect to $G$. Since strict equivariance can be overly restrictive in many scenarios, relaxing equivariance to an appropriate extent has become a more practical alternative. We revisit equivariance relaxation from the perspective of quantifying equivariance error. To this end, we adopt a quantitative formulation of equivariance introduced in Fei & Deng (2024); Wang et al. (2022):

**Definition 4.** *(Equivariance quantification)* A function $f : \mathcal{X} \to \mathcal{Y}$ is said to be C-weakly equivariant with respect to a group $G$ if it satisfies: $\mathcal{E}(f) = \mathbb{E}_{x,g}[\|f(\rho(g)x) - \rho(g)f(x)\|] \leq C$, where $g$ is sampled from the group $G$ according to the normalized Haar measure $\mu$.

A smaller value of $\mathcal{E}(f)$ indicates stronger equivariance of the function $f$ with respect to the group $G$, and vice versa. Therefore, if a method is said to implement equivariance relaxation, it implies that its equivariant error $\mathcal{E}(f)$ is relatively large or appropriately increased compared to the strictly equivariant counterpart.

# 4 METHOD

This section presents the three components of our method, as illustrated in the pipeline (Fig. 6).

## 4.1 RESIDUAL PATHWAY FOR VECTOR NEURONS

Inspired by ResNet and RPP, we propose a residual connection architecture for VN: $f(x) = \Phi(x) + \Psi(\Phi(x))$ (see Fig.1). The core idea is to view the overall model as the sum of an equivariant function $\Phi$ with a symmetric inductive bias enforced by rigid structural constraints and a correction function $\Psi$ that can be flexibly adjusted. The $\Phi$ can directly utilize VNNs (e.g., VN-PointNet, VN-DGCNN, VN-T, etc.). This residual architecture tends to use the more structured function $\Phi$ to interpret the data, and, when the data is more complex, to employ the function $\Psi$ to explain the discrepancies between the target and the part of the data already captured by the function $\Phi$. This naturally transforms the rigid inductive bias into a soft prior, thereby enabling VNNs to break equivariance.

Our residual connection architecture leverages the inherent advantage of the VN framework. Since both the input and output of the VNNs lie in a 3D manifold, the group action $\rho(g)$ has the same form in both the input and the representation spaces. Consequently, for correction functions $\Psi$, enforcing equivariance on the representation $\Phi(x)$ is equivalent to enforcing it on the input $x$. In contrast, Other equivariant approaches do not share this property, making it difficult to flexibly adjust equivariance constraints within a residual connection paradigm.

## 4.2 IMPLICIT EQUIVARIANCE REGULARIZATION

In this section, we derive a mechanism for adaptive equivariance from a general objective function for equivariant tasks. Our theoretical analysis demonstrates that by minimizing this objective, even non-equivariant models can acquire equivariance that adapts to the intrinsic symmetry of the data. Our Ada-VNNs achieve adaptive equivariance by leveraging this mechanism.

Let $f : \mathcal{X} \to \mathcal{Y}$ be a neural network mapping from the input space $\mathcal{X}$ to the output space $\mathcal{Y}$. Both spaces are equipped with a group action by $G$, denoted by the representation $\rho$. In such tasks, the goal is to minimize the error between the predicted output and the ground-truth output under transformed inputs. The general objective can be written as:

$$\mathcal{L} = \mathbb{E}_{x,g}\left[\|f\left(\rho(g)x\right) - \rho(g)y\|\right] \tag{1}$$

where $x, y$ are sampled from the data distribution $\mathcal{D}$, $g$ is sampled from the SO(3) group $G$ according to the normalized Haar measure $\mu$, and the loss metric is the $L_2$ norm.

**Assumption 1.** (Intrinsic Symmetry Breaking Wang et al. (2022)). We assume the data pairs $\{x, y\}$ are generated by a ground-truth physical process $f^*$ (i.e., $y = f^*(x)$). We quantify the inherent symmetry breaking of $f^*$ using the supremum $\epsilon$:

$$\sup_{x,g} \| f^*(\rho(g)x) - \rho(g)f^*(x) \| = \epsilon \tag{2}$$

Here, $\epsilon$ represents the intrinsic deviation of the ground truth from strict equivariance. $\epsilon = 0$ indicates perfect symmetry, while $\epsilon > 0$ indicates symmetry breaking.

**Proposition 1 (Adaptive Equivariance Bound).** *Under Assumption 1, and assuming the training distribution covers the group orbit (i.e., via data augmentation), the equivariance error $\mathcal{E}(f) = \mathbb{E}_{x,g}[\| f(\rho(g)x) - \rho(g)f(x) \|]$ of the learned function $f$ is bounded by the task loss $\mathcal{L}$ and the intrinsic symmetry breaking $\epsilon$:*

$$\mathcal{E}(f) \leq 2\mathcal{L} + \epsilon \tag{3}$$

*Proof.* First, using the triangle inequality, we decompose the equivariance error $\mathcal{E}(f)$:

$$\mathcal{E}(f) = \mathbb{E}_{x,g}[\| f(\rho(g)x) - \rho(g)f(x) \|]$$
$$\leq \underbrace{\mathbb{E}_{x,g}[\| f(\rho(g)x) - \rho(g)f^*(x) \|]}_{\mathcal{L}} + \underbrace{\mathbb{E}_{x,g}[\| \rho(g)f^*(x) - \rho(g)f(x) \|]}_{\text{Residual}} \tag{4}$$

Since $\rho(g) \in$ SO(3) is norm-preserving, the residual term simplifies to $\mathbb{E}_x[\| f^*(x) - f(x) \|]$. Thus:

$$\mathcal{E}(f) \leq \mathcal{L} + \mathbb{E}_x[\| f^*(x) - f(x) \|] \tag{5}$$

Next, we relate the residual term $\mathbb{E}_x[\| f^*(x) - f(x) \|]$ to the loss $\mathcal{L}$. Using the reverse triangle inequality variation $\| a - b \| \geq \| a - c \| - \| c - b \|$:

$$\mathcal{L} = \mathbb{E}_{x,g}[\| f(\rho(g)x) - \rho(g)f^*(x) \|]$$
$$\geq \mathbb{E}_{x,g}[\| f(\rho(g)x) - f^*(\rho(g)x) \|] - \mathbb{E}_{x,g}[\| f^*(\rho(g)x) - \rho(g)f^*(x) \|] \tag{6}$$

We apply two properties here: 1) From *Assumption 1*, the constraint $\| f^*(\rho(g)x) - \rho(g)f^*(x) \| \leq \epsilon$ holds for all $x, g$. 2) For the first term, let $z = \rho(g)x$. We utilize the Volume Invariance of $\rho(g)$ ($| \det(\rho) | = 1$) to preserve the integration measure ($dz = dx$). Combined with the standard assumption that the training distribution covers the group orbit (achieved via augmentation), the expectation remains consistent: $\mathbb{E}_{x,g}[\| f(\rho(g)x) - f^*(\rho(g)x) \|] = \mathbb{E}_x[\| f(x) - f^*(x) \|]$.

Substituting these into the inequality, we obtain a lower bound for $\mathcal{L}$:

$$\mathcal{L} \geq \mathbb{E}_x[\| f(x) - f^*(x) \|] - \epsilon \tag{7}$$

Finally, substituting this result back into Eq. (5), we obtain $\mathcal{E}(f) \leq 2\mathcal{L} + \epsilon$. $\qquad\square$

**Application to Ada-VNN Architecture.** Considering our proposed residual architecture $f(x) = \Phi(x) + \Psi(\Phi(x))$, where $\Phi(x)$ is strictly equivariant by design (i.e., $\Phi(\rho(g)x) = \rho(g)\Phi(x)$). We show the total equivariance error $\mathcal{E}(f)$ is determined solely by the residual branch $\Psi$:

$$\| f(\rho(g)x) - \rho(g)f(x) \| = \| [\rho(g)\Phi(x) + \Psi(\rho(g)\Phi(x))] - [\rho(g)\Phi(x) + \rho(g)\Psi(\Phi(x))] \|$$
$$= \| \Psi(\rho(g)\Phi(x)) - \rho(g)\Psi(\Phi(x)) \| \tag{8}$$

Thus, $\mathcal{E}(f) = \mathcal{E}(\Psi)$. Applying the bound from Proposition 1:

$$\mathcal{E}(f) = \mathcal{E}(\Psi) \leq 2\mathcal{L} + \epsilon \tag{9}$$

Here, $\mathcal{E}(\Psi)$ denotes the equivariance error of the residual branch $\Psi$ evaluated over the latent feature distribution. Eq. 9 reveals the mechanism of implicit equivariance regularization: minimizing the prediction task loss $\mathcal{L}$ implicitly constrains the equivariance error of $\Psi$.

The behavior of the model depends on the intrinsic property $\epsilon$:

**Case 1: No Symmetry Breaking** ($\text{Stab}_G(y) \supseteq \text{Stab}_G(x)$, $\epsilon \to 0$). When the underlying physics is perfectly symmetric, the bound becomes $\mathcal{E}(f) \leq 2\mathcal{L}$. As the network minimizes the task loss

($\mathcal{L} \to 0$), the equivariance error is forced to vanish ($\mathcal{E}(f) \to 0$). The model converges to a strictly equivariant function.

**Case 2: Symmetry Breaking** ($\text{Stab}_G(y) \not\supseteq \text{Stab}_G(x)$, $\epsilon > 0$). When inherent symmetry breaking exists, the bound becomes $\mathcal{E}(f) \leq 2\mathcal{L} + \epsilon$. Minimizing $\mathcal{L}$ does not enforce zero equivariance error but restricts it within a margin of $\epsilon$. The model converges to an $\epsilon$-weakly equivariant state.'

This theoretically proves that Ada-VNN achieves adaptive equivariance: the strictness of the equivariance constraint is dynamically determined by the intrinsic symmetry strength of the data, rather than being manually tuned. This naturally realizes a subtle form of adaptive equivariance.

### 4.3 LIGHTWEIGHT DESIGN OF THE CORRECTION FUNCTION

Equivariance is easily broken but difficult to establish. In the VN framework, establishing equivariance requires constraining all layers of the model, whereas breaking it requires only a single linear layer to violate the equivariance constraint. Therefore, models with tunable equivariance should be kept as compact as possible, as a larger number of parameters makes enforcing equivariance more difficult. Based on this insight, we propose a lightweight and interpretable DV-MLP as the structural form of the correction function $\Psi$.

Classical VNNs achieve SO(3)-equivariance by omitting bias terms, as biases introduce fixed directional components that break rotational symmetry. The core of the Vector Linear Layer (VLL) lies in applying a shared weight matrix across the $x$, $y$, and $z$ directions, ensuring consistent transformations for all vector neurons:

$$\text{VLL}(V) = [WV^x, WV^y, WV^z],\qquad(10)$$

here, $W \in \mathbb{R}^{C' \times C}$ is the shared weight matrix, and $V^x, V^y, V^z \in \mathbb{R}^{C \times 1}$ represent the channels of the input vector along the $x$, $y$, and $z$ directions, respectively. This setup naturally preserves equivariance in 3D space. We propose the Decoupled Vector Linear Layer (DVLL) (Fig. 7), which discards the weight-sharing strategy and uses three distinct weight matrices to independently transform the $V^x$, $V^y$, and $V^z$ channels of each vector neuron:

$$\text{DVLL}(V) = \left[W^1 V^x, W^2 V^y, W^3 V^z\right]\qquad(11)$$

This approach explicitly breaks equivariance. Crucially, DVLL provides a fully controllable and interpretable mechanism for equivariance adjustment: when the three submatrices are identical, DVLL reduces to a standard VLL and thus preserves strict equivariance; the degree of deviation among the submatrices quantitatively reflects the extent to which equivariance is broken. We provide more details in the Appendix Sec. 9.2.

DV-MLP is built upon the proposed DVLL and employs VN-ReLU—an equivariant variant of ReLU introduced in VNN Deng et al. (2021)—as its nonlinear layer: DV-MLP = $[\text{VN-ReLU} \circ \text{DVLL}]^k$, here, $k$ represents the number of hidden layers. For the equivariant representation $\Phi(x) \in \mathbb{R}^{C \times 3}$, the output representation is given by: $f(x) = \Phi(x) + \text{DV-MLP}(\Phi(x)) \in \mathbb{R}^{C' \times 3}$.

## 5 EXPERIMENT

Our experiments are designed to answer the following questions: (1) Can Ada-VNNs achieve adaptive equivariance? (Sec. 5.1) (2) Does the optimization of the objective function $\mathcal{L}$ implicitly minimize the model's equivariance error during training? (Sec. 5.2) (3) How do Ada-VNNs perform on equivariant task compared to the original VNNs? Are they able to effectively handle highly symmetric inputs? (Sec. 5.3) (4) Why does our proposed approach outperform strictly equivariant or fully non-equivariant baselines? (Sec. 5.4) (5) How do the number of residual connections and the depth of DV-MLP affect equivariance and overall performance? (Sec. 5.5)

### 5.1 ADAPTIVE EQUIVARIANCE

**Task and Data**. Pose estimation is a fundamental 3D task that is strictly SO(3)-equivariant when translation is ignored. Following VNN, EPN, and E2PN, the model takes a pair of point clouds as input and predicts their relative rotation, with all methods using the same MLP decoder to regress

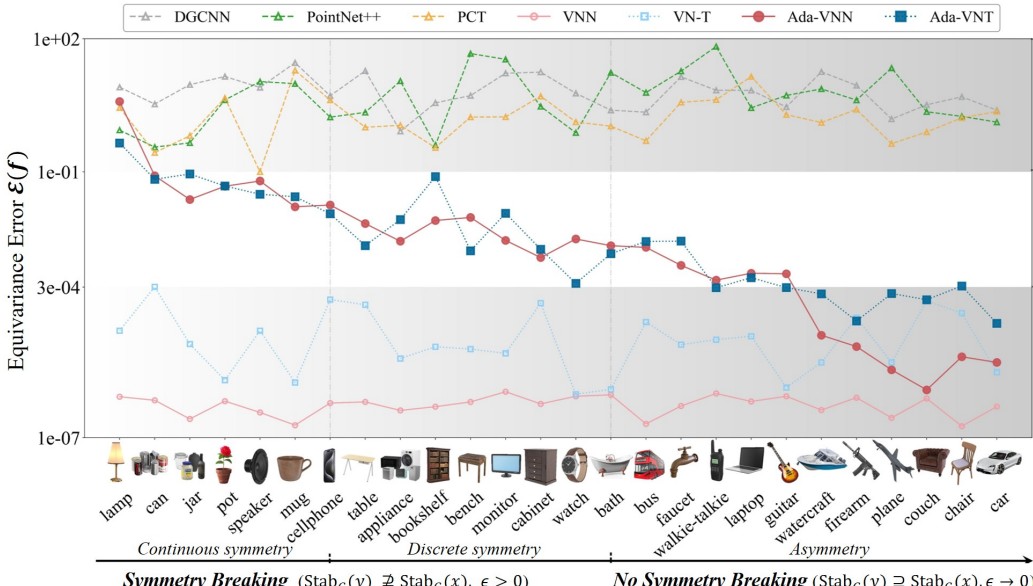

Figure 2: Experiments with 7 models across 26 categories with different input symmetries show that only Ada-VNNs achieve adaptive equivariance. The non-equivariant threshold is set as the lowest error among non-equivariant models (PCT on speaker, 1e-1), and the equivariant threshold is set as the highest error among strongly equivariant models (VN-T on can, 3e-4).

rotation parameters. We evaluate on 26 ShapeNet categories (Fig. 2): six (lamp–mug) with continuous rotational symmetry $G_s \cong \mathrm{SO}(2)$, nine (cellphone–watch) with discrete rotational symmetry $G_s \cong C_n, n \geq 2$, and eleven (bus–car) that are largely asymmetric. In pose estimation task, categories with continuous or discrete symmetries are both subject to symmetry breaking.

**Metric.** We adopt a discretized version of the equivariance quantification formula (Def. 4) to measure model equivariance. Lower equivariance error indicates stronger equivariance.

**Baseline**. We compared a range of point cloud networks with different degrees of equivariance: (1) Non-equivariant networks: PointNet++, DGCNN, and PCT. (2) VN-based counterparts: VNN and VN-T. (3) Adaptive counterparts: Ada-VNN and Ada-VNT. Each of the above models was trained separately on 26 categories with distinct symmetries. Subsequently, we discarded the decoder and computed the equivariance error represented by each model. Further expeirments details can be found in Appendix Sec. 9.3.

**Result.** As shown in Fig. 2, for non-equivariant networks (PointNet++, DGCNN, PCT), it is difficult to establish equivariance under a task-specific loss function. Regardless of the input category's symmetry type, the representations learned by these models cannot be regularized, resulting in consistently large equivariance errors and poor equivariance performance. In contrast, equivariant networks such as VNN and VN-T consistently maintain strong equivariance across all categories, indicating that they are unable to actively break equivariance constraints.

However, Ada-VNNs exhibit behavior fundamentally different from both equivariant and non-equivariant paradigms. The equivariance performance of Ada-VNN and Ada-VNT shows a clear negative correlation with the symmetry level of the training data. When the symmetry of the training categories is high (i.e.,$\mathrm{Stab}_G(y) \not\supseteq \mathrm{Stab}_G(x), \epsilon = 0$), the model is able to actively break the equivariance constraints, thereby exhibiting weak equivariance. In contrast, when the training categories are asymmetry (i.e., $\mathrm{Stab}_G(y) \supseteq \mathrm{Stab}_G(x), \epsilon \to 0$), the model maintains strong equivariance. This indicates that Ada-VNNs allow the model to relax equivariance constraints according to input symmetry. The observed trend suggests that the model's equivariance emerges from statistical optimization over the symmetry present in the training data. Rather than belonging to strict categories such as "equivariant" or "non-equivariant," the model gradually adjusts its behavior along a spectrum defined by the degree of symmetry.

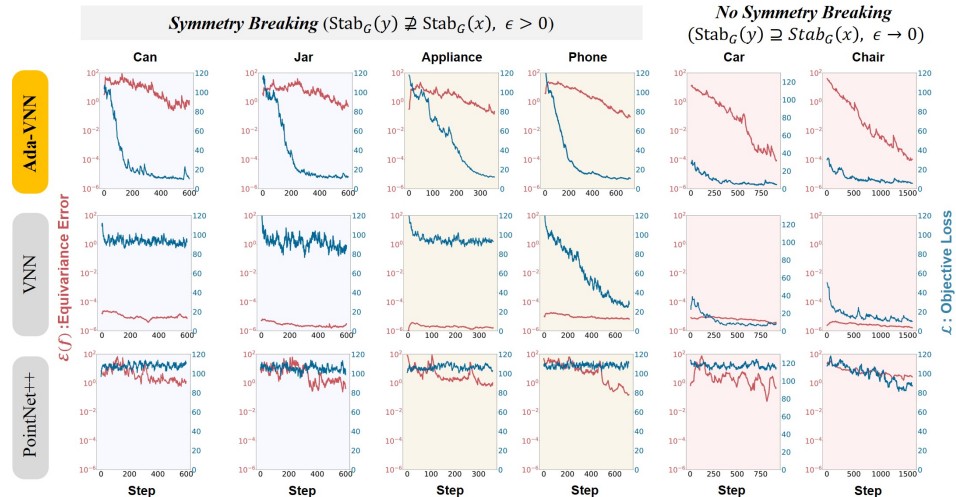

Figure 3: Training Dynamics: Equivariance Error vs. Optimization Objective. We visualize the change in both the optimization objective $\mathcal{L}$ and the equivariance error $\mathcal{E}(f)$ over the training steps.

## 5.2 VERIFICATION OF IMPLICIT REGULARIZATION

For further analysis, we visualize the evolution of the equivariance error and the optimization objective for Ada-VNN, VNN, and PointNet++ across different symmetry categories during training.

As shown in Fig. 3, PointNet++, VNN, and Ada-VNN exhibit three distinct behavioral patterns. VNN enforces equivariance through predefined structural constraints, which maintain its equivariant properties consistently during training, largely unaffected by input symmetry. In contrast, PointNet++ shows a slight decrease in equivariance error over training.

Ada-VNNs exhibit distinct variations in equivariance error throughout the training process. Specifically, when the degree of symmetry breaking in the data is significant (e.g., the 'Can' and 'Jar' categories in the figure), corresponding to a large $\epsilon$, the model's equivariance error $\mathcal{E}(f)$ remains relatively high even after the objective function $\mathcal{L}$ converges to a small value. As the degree of symmetry breaking decreases (e.g., the 'Appliance' and 'Phone' categories), $\epsilon$ gradually diminishes, leading to a further reduction in the model's equivariance error upon the convergence of $\mathcal{L}$. Finally, when the data exhibits minimal symmetry (e.g., the 'Car' and 'Chair' categories), where $\epsilon \to 0$, the equivariance error is effectively suppressed to approximately $10^{-4}$ following the convergence of $\mathcal{L}$. This observation supports our reasoning on implicit equivariance regularization (Sec. 4.2 ).

## 5.3 ROBUSTNESS TO SYMMETRY BREAKING

In this section, we demonstrate the significant impact of the adaptive equivariant properties of Ada-VNNs on pose estimation performance, particularly for categories with high symmetry.

**Result**. As shown in Tab. 1, equivariant models like VNN and VN-T perform poorly on categories with continuous rotational symmetry. This is especially evident for VNN, which nearly collapses on lamp, can, jar, pot, and speaker, with recall dropping below 5%. Non-equivariant models like PointNet++ and PCT also underperform across most categories.

In contrast, the proposed Ada-VNN and Ada-VNT show substantial gains on highly symmetric categories. Ada-VNN improves average performance by 53 points on continuous and discrete classes, and by up to $33\times$ on the highly symmetric lamp category (see Fig. 4 (right)). Ada-VNT achieves better results across symmetric categories while maintaining strong performance on asymmetric ones. These findings suggest that adaptive models effectively mitigate symmetry breaking issues and deliver balanced performance across varying symmetry levels. Fig. 4 (left) presents qualitative pose estimation results of VNN and Ada-VNN under inputs with varying degrees of symmetry. For highly symmetric inputs, VNN fails to perform accurate pose estimation, leading to mismatches,

Table 1: Comparison results on 26 Categories. Registration recall at a 10° threshold is used as the performance metric; higher is better. **Bold:** best, underline: second best.

| Method | Continuous symmetry, $\epsilon > 0$ | | | | | | | Discrete symmetry, $\epsilon > 0$ | | | | | |
|---|---|---|---|---|---|---|---|---|---|---|---|---|---|
| | lamp | can | pot | jar | speaker | mug | phone | table | appliance | bookshelf | cabinet | monitor | bench |
| PointNet++ | 0.00 | 0.71 | 0.00 | 0.00 | 0.00 | 0.00 | 0.00 | 0.00 | 0.00 | 0.70 | 0.00 | 0.00 | 0.00 |
| PCT | 0.00 | 0.00 | 0.00 | 0.00 | 0.00 | 0.00 | 0.00 | 7.30 | 0.00 | 0.00 | 0.00 | 8.91 | 7.86 |
| EPN | 90.98 | 70.00 | 5.98 | 5.50 | 84.87 | 6.20 | 83.83 | 94.49 | 3.83 | 30.00 | 89.67 | **84.59** | 80.31 |
| VNN | 2.38 | 4.29 | 1.11 | 4.51 | 3.44 | 37.50 | 12.86 | 49.53 | 2.33 | 14.79 | 15.87 | 18.72 | 50.69 |
| **Ada-VNN** | 79.44 | 75.71 | 55.56 | 62.41 | 76.88 | **71.88** | 70.48 | 93.19 | 35.58 | 71.13 | 70.79 | 73.52 | 86.78 |
| | *+77.06* | *+71.42* | *+54.45* | *+57.90* | *+73.44* | *+34.38* | *+57.62* | *+43.66* | *+23.25* | *+56.34* | *+54.92* | *+54.80* | *+36.09* |
| VN-T | 79.39 | 53.57 | 34.44 | 57.89 | 75.50 | 6.25 | 76.67 | 97.33 | 45.35 | 71.83 | 81.30 | 72.60 | **92.84** |
| **Ada-VNT** | **92.93** | **77.71** | **57.33** | **67.01** | **85.44** | 52.13 | 83.52 | **98.21** | **56.33** | **76.20** | 87.06 | 82.34 | 90.23 |
| | *+13.54* | *+24.14* | *+22.89* | *+9.12* | *+9.94* | *+45.88* | *+6.85* | *+0.88* | *+10.98* | *+4.37* | *+5.76* | *+6.74* | *-2.61* |

| Method | Asymmetry, $\epsilon \to 0$ | | | | | | | | | | | Discrete symmetry | |
|---|---|---|---|---|---|---|---|---|---|---|---|---|---|
| | bus | laptop | guitar | faucet | walkie-talkie | watercraft | firearm | couch | chair | airplane | car | bath | watch |
| PointNet++ | 0.00 | 1.45 | 0.00 | 0.00 | 0.00 | 0.00 | 0.00 | 0.00 | 0.03 | 0.00 | 0.00 | 0.00 | 0.00 |
| PCT | 0.00 | 2.10 | 1.67 | 3.80 | 7.61 | 4.52 | 0.00 | 0.00 | 8.59 | 10.25 | 0.00 | 0.00 | 0.00 |
| EPN | 43.52 | 38.90 | 88.54 | 49.91 | 42.82 | **93.16** | 85.50 | 86.10 | 92.89 | 94.25 | 95.22 | **93.40** | **93.14** |
| VNN | 52.86 | 49.28 | 89.92 | 22.52 | 13.71 | 66.75 | 84.84 | 88.35 | 92.85 | 80.72 | **97.30** | 22.66 | 4.12 |
| **Ada-VNN** | **85.00** | 84.06 | 94.96 | 50.45 | **70.16** | 89.69 | 92.84 | 91.65 | 96.46 | **96.17** | 97.16 | 73.44 | 58.76 |
| | *+32.14* | *+34.78* | *+5.04* | *+27.93* | *+56.45* | *+22.94* | *+8.00* | *+3.30* | *+3.61* | *+15.45* | *-0.14* | *+50.78* | *+54.64* |
| VN-T | 72.86 | **91.30** | 93.96 | **72.97** | 58.06 | 86.86 | 90.74 | **93.54** | 96.76 | 93.20 | 96.87 | 78.91 | 68.04 |
| **Ada-VNT** | 75.29 | 88.26 | **95.80** | 69.46 | 54.84 | 88.40 | 91.45 | 91.97 | **99.19** | 93.70 | 95.16 | 80.88 | 69.67 |
| | *+2.43* | *-3.04* | *+1.84* | *-3.51* | *-3.22* | *+2.00* | *+0.00* | *-1.57* | *+2.43* | *+0.50* | *-1.71* | *+1.97* | *+1.63* |

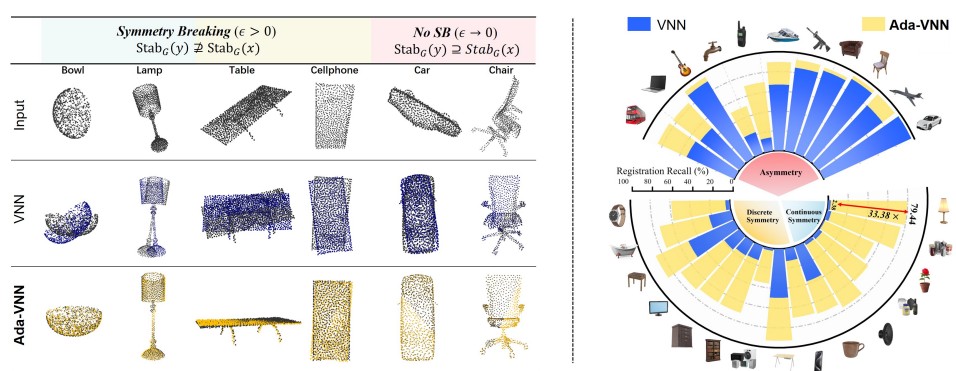

Figure 4: Qualitative (left) and quantitative (right) results across varying symmetries.

whereas Ada-VNN performs robustly across different levels of symmetry. Continuous symmetries are particularly challenging, yet Ada-VNN is surprisingly able to achieve alignment, indicating its ability to extract preferred low-symmetry outputs from highly symmetric inputs. This property shows strong potential for broader near-symmetric tasks, such as opening cans or lockers.

## 5.4 WHY ADA-VNNs WORK?

In this section, we aim to explain why Ada-VNNs outperform strictly equivariant and fully non-equivariant methods on tasks involving symmetry breaking.

**Feature Collapse .** For strictly equivariant methods, feature collapse occurs when real-world data possesses approximate symmetry (e.g., objects such as cans or lamps). This strong equivariance constraint leads to performance degradation in tasks like rotation registration. Assume the input $x$ is approximately symmetric with respect to the group $G_s$; that is, for any $g_s \in G_s$, we have $\rho(g_s)x \approx x$. We define the deviation as $\delta = \|\rho(g_s)x - x\|$. For a strictly equivariant function $f$, the condition $f(\rho(g_s)x) = \rho(g_s)f(x)$ must hold. Leveraging $k$-Lipschitz continuity, we obtain:

$$\|\rho(g_s)f(x) - f(x)\| = \|f(\rho(g_s)x) - f(x)\| \leq k\|\rho(g_s)x - x\| = k\delta \tag{12}$$

The term on the left, $\|\rho(g_s)f(x) - f(x)\| \leq k\delta$, implies that when the input is nearly symmetric ($\delta \to 0$), the output feature $f(x)$ is forced to approach the Fixed Point Set of the group $G_s$. Consequently, if the group $G_s$ represents continuous rotational symmetry about a specific axis, the components of $f(x)$ lying in the plane orthogonal to the rotation axis are forced to approach zero. This results in the

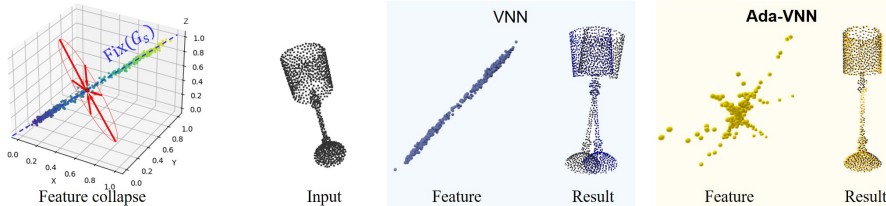

Figure 5: Feature collapse phenomenon and comparative visualization. VNN's features collapse to the Fixed Point Set, while Ada-VNN avoids collapse and achieves robust registration. We present further visualizations in Figure 11.

loss of critical geometric information required to distinguish orientation, as visualized in Figure 5. In contrast, Ada-VNN allows the equivariance error to retain a deviation within the margin $\epsilon$. This flexibility allows the model to leverage critical asymmetric features to resolve ambiguity.

**Superiority over Non-Equivariant Methods.** Fully non-equivariant methods face two main challenges. First, these methods lack rotation inductive bias, which makes it difficult for them to generalize to rotation angles unseen during training. Second, it is inherently challenging to perform point cloud registration directly from features generated by a non-equivariant encoder without task-specific design. Prior methods that successfully applied backbones like PointNet++ or PointTransformer to registration tasks Yang et al. (2024a) all relied on auxiliary modules, such as feature matching, specifically tailored for the task.

## 5.5 ABLATION STUDY

In this section, we investigate the contribution of the non-equivariant layers (DV-MLP) and the depth of the residual connections ($\Psi$) to Ada-VNN's performance and adaptive behavior.

**Effect of DV-MLP.** To study the performance-equivariance trade-off, we ablate the DV-MLP depth ($k = 0, 1,$ and $2$). As shown in Fig. 9, the DV-MLP significantly improves VNN performance (up to a $33\times$ boost on symmetric categories like 'Lamp'). However, this gain is costly: each added layer increases the equivariance error by nearly an order of magnitude. This error rapidly grows and saturates performance, demonstrating high sensitivity to non-equivariant parameters. This effective control over non-equivariant parameter counts explains Ada-VNNs' superior adaptation compared to non-equivariant baselines.

As shown in Fig. 10(b) and Tab. 2, the residual correction function $\Psi$ is highly effective and lightweight. A single residual connection is sufficient to yield substantial performance gains (up to $508\%$ avg. improvement over VNN), yet it only introduces a marginal memory overhead of $0.2\%$. This confirms that Ada-VNN's success is rooted in a substantial architectural insight for symmetry breaking, rather than relying on parameter stacking.

**Effect of residual connections.** We ablated the impact of residual layer count ($l = 1, 2, 3$). As shown in Fig. 10(a), Ada-VNN's equivariance is influenced by data symmetry, confirming its adaptivity even with a single correction function $\Psi$. However, increasing residual depth has a drastic impact: compared to adding DV-MLP layers, each additional layer increases the equivariance error by approximately $10^4$.

## 6 CONCLUSION

In this work, we analyzed the fundamental limitations of VNNs in handling symmetry breaking. To address these issues, we proposed Ada-VNNs for flexible relaxation of equivariance constraints, offering a first step toward better understanding how VNNs handle symmetry breaking on SO(3). There remain many avenues for extending this work. First, further validation is needed across a broader range of tasks in different domains. Second, it will be important to explore how our approach can be applied to other equivariant networks (such as frame averaging). Finally, flexible relaxation methods could be further discussed in the context of the E(3) and SIM(3) groups to adapt to diverse application scenarios.

## 7 ETHICAL STATEMENT

We have read and acknowledge the ICLR Code of Ethics https://iclr.cc/public/CodeOfEthics and comply with its guidelines. Our paper contains no content that violates ethical or moral standards.

## 8 REPRODUCIBILITY STATEMENT

We provide detailed derivations of the formulas in Appendix Sec. **??** and have included a link to the anonymous code repository https://github.com/anon-mity/Ada-VNNs. This will assist researchers in the industry in replicating our experiments and results.

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

# 9 APPENDIX

## 9.1 THE PIPELINE OF ADA-VNNS.

In this section, we present our approach in a more intuitive manner. As illustrated in Fig. 6, if we view an elastic rope as an analogy for an equivariant model, then rigid equivariance constraints can be thought of as nails embedded in the rope.

1. The residual architecture acts like removing the nails: it converts the rigid inductive bias of equivariance constraints into a soft prior, thereby providing VNNs with the fundamental ability to break equivariance.

2. Implicit equivariance regularization acts like stretching the rope: it adaptively adjusts the strength of equivariance constraints according to the degree of symmetry in the input data.

3. DV-MLP serves to make the rope lightweight and interpretable: by reducing parameter complexity, it allows the equivariance constraints to be more easily tuned during training. Moreover, the relaxation degree is explicitly reflected in the distances between linear layers. Together, these components yield Ada-VNNs.

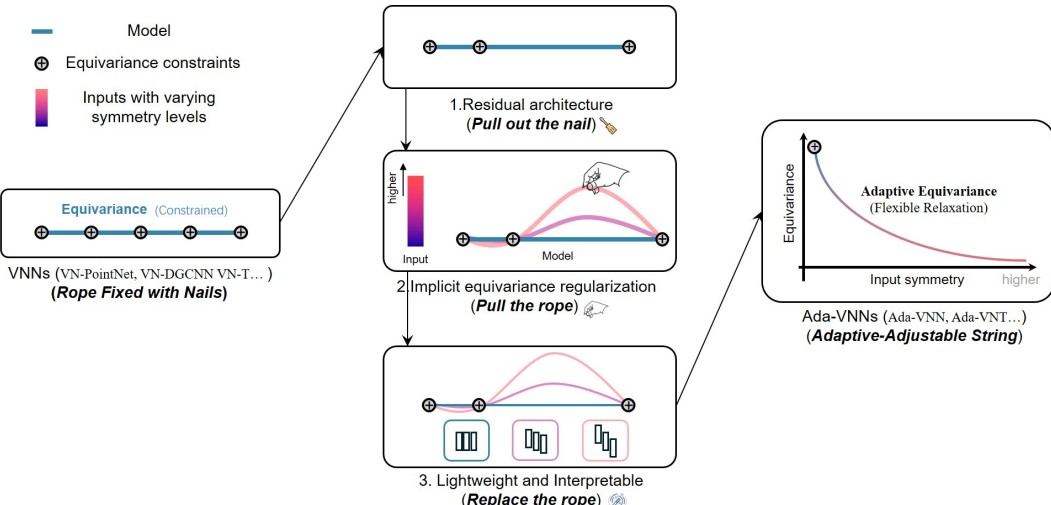

Figure 6: Visual analogy illustrating the evolution to Ada-VNNs: Starting from equivariant neural networks (VNNs, represented as a "rope fixed with nails" to denote rigid equivariance constraints), the diagram depicts three key components: (1) residual architecture (analogized to "pulling out nails" to relax rigid constraints), (2) implicit equivariance regularization (analogized to "pulling the rope" to adapt the symmetry level of inputs), and (3) lightweight/interpretable design via DV-MLP (analogized to "replacing the rope" for efficiency and interpretability). These culminate in the "adaptive-adjustable string" paradigm of Ada-VNNs, where equivariance flexibly relaxes with input symmetry.

## 9.2 MORE DETAILS ON DV-MLP

In this section, we prove that the equivariance of DV-MLP is tunable and can be mapped via the distance between linear weight matrices. We begin by introducing the mechanism of the Vector Linear Layer (VLL), which is conceptually similar to a scalar linear layer, except that it applies the same scalar transformation across the three spatial dimensions. For an input vector $V \in \mathbb{R}^{C \times 3}$, VLL learns a weight matrix $W \in \mathbb{R}^{C' \times C}$, and the transformation is given by:

$$V' = \text{VLL}(V) = [WV^x, WV^y, WV^z] \tag{13}$$

$$V'_{c',i} = \sum_{c=1}^{C} W_{c',c} V_{c,i} \in \mathbb{R}^{C' \times 3} \tag{14}$$

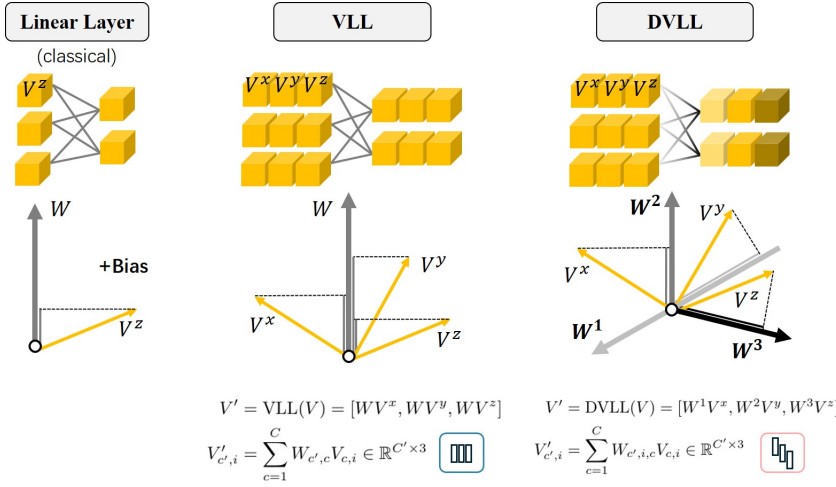

Figure 7: Different configurations of scalar, vector, and decoupled vector linear layers.

VLL shares the same linear transformation across the three spatial channels, preserving equivariance. In contrast, DVLL removes this weight sharing and instead uses three separate weight matrices $W^1, W^2, W^3$ to learn directional features independently, as expressed by:

$$V' = \text{DVLL}(V) = [W^1 V^x, W^2 V^y, W^3 V^z] \tag{15}$$

$$V'_{c',i} = \sum_{c=1}^{C} W_{c',i,c} V_{c,i} \in \mathbb{R}^{C' \times 3} \tag{16}$$

Here, $c'$, $i$, and $c$ correspond to output channel, spatial dimension, and input channel, respectively.

Next, we analyze how the distances among the independent weight matrices in DVLL relate to equivariance breaking. To quantify this, we define the rotation-induced deviation:

$$\mathcal{E}_{\text{dev}}(V, R) := \|\text{DVLL}(VR^\top) - \text{DVLL}(V)R^\top\|_F \tag{17}$$

where $V \in \mathbb{R}^{C \times 3}$ is the input, $R \in \text{SO}(3)$ is a spatial rotation matrix, and $\|\cdot\|_F$ denotes the Frobenius norm; clearly, if $W^1 = W^2 = W^3$, DVLL reduces to VLL and $\mathcal{E}_{\text{dev}} = 0$, indicating perfect equivariance.

We define the total directional weight deviation as:

$$\Delta := \|W^1 - W^2\|_F^2 + \|W^1 - W^3\|_F^2 + \|W^2 - W^3\|_F^2 \tag{18}$$

which measures the overall inconsistency among the three directional weight matrices. Clearly, $\Delta = 0$ if and only if $W^1 = W^2 = W^3$, in which case DVLL reduces to VLL and achieves perfect equivariance. As $\Delta$ increases, directional asymmetry is introduced, correlating with larger equivariance deviations. While the exact relationship between $\Delta$ and $\mathcal{E}_{\text{dev}}$ depends on the input, $\Delta$ provides a practical and tunable measure of weight asymmetry, allowing controlled relaxation of equivariance in DV-MLP. In other words, adjusting $\Delta$ modulates the degree of symmetry breaking in a principled and interpretable manner.

## 9.3 EXPERIMENTAL DETAILS

**Data** AtlasNetH5Katzir et al. (2022); Kim et al. (2023), a subset of ShapeNet Chang et al. (2015), contains point clouds from 13 categories. To enable more comprehensive evaluation, we extend the dataset with 13 additional categories. The point clouds for these additional categories are generated by applying surface farthest point sampling to the 3D mesh models (OBJ files) from ShapeNet-Corev2. Categories with insufficient data in ShapeNet are excluded, as they are inadequate to support reliable experiments. In total, the resulting dataset used for evaluation comprises 26 object categories, with 39,309 shapes used for training and 9,834 shapes for testing, following an 80:20 split.

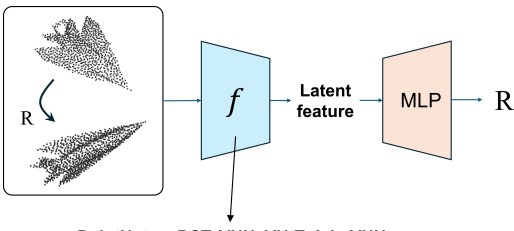

Figure 8: Framework for comparative experiments.

**Training** Each point cloud contains 1024 points. A training pair is formed by applying a random rotation to a ShapeNet point cloud, and the model is tasked with predicting the relative rotation between the pair. All models are trained under a unified protocol with a batch size of 32, the Adam optimizer (initial learning rate 0.001), and Geodesic Distance (GD) as the loss function for 200 epochs. To ensure a fair comparison, we avoid using any specially designed decoder and instead adopt a simple MLP to directly regress the rotation from the learned representation (see Fig. 8 ). Regarding the role of the decoder in the pose estimation task, we strictly followed the experimental settings established in EPN and E2PN. Specifically, the proposed model functions as a feature extractor to derive representations from the input point cloud, followed by a simple MLP acting as a decoder to directly regress rotation parameters. The rationale behind this setup is to assess whether the model's representations are beneficial for the downstream pose estimation task by comparing performance. This approach is analogous to the linear probing protocol, aiming to provide a fair comparison of representation quality derived from different backbones.

This setup is analogous to the linear probing protocol He et al. (2022) widely used for evaluating pretrained representations, as it isolates the representational quality of the model from the complexity of the decoder.

**Metric** we using the discretized version of the equivariance quantization formula presented in Definition 4:

$$\mathcal{E}(f)_{\text{discrete}} = \frac{1}{|X_{\text{test}}| \cdot N} \sum_{x \in X_{\text{test}}} \sum_{i=1}^{N} \text{CD}\big(f(R_i x),\, R_i f(x)\big) \tag{19}$$

For each shape in the test set $X_{\text{test}}$, we generate a rotated counterpart by applying a rotation $R_i$ sampled from a set of random rotational transformations, resulting in a set of point cloud pairs $\{x, R_i x\}_{i=1}^{N}$, where $N = 120$. We then compute the two-way Chamfer Distance (CD) between the transformed representation $R_i f(x)$ and the representation of the rotated input $f(R_i x)$. In theory, smaller distances indicate stronger equivariance in the learned representations.

From a performance perspective, VNN struggles with symmetric inputs (e.g., Can, Jar, Appliance), where training process oscillates and fails to converge. In contrast, VNN quickly converges on asymmetric inputs. PointNet++ performs worse overall, with only marginal success on the "Chair" class. This is because PointNet++ lacks equivariant features aligned with output, and predicting pose directly from unstructured MLPs without specialized decoders is inherently difficult. Only AdaR-VNN performs well across all symmetry types. This indicates that Ada-VNN achieves a flexible balance between the model's equivariance constraints and task performance.

## 9.4 ABLATION STUDY

## 9.5 THE USE OF LARGE LANGUAGE MODELS (LLMS)

LLMs did not play a significant role in the conception, writing, or any other part of this work. Therefore, LLMs are not considered contributors. The only use of LLMs in this work was for checking spelling or grammatical errors in the writing.

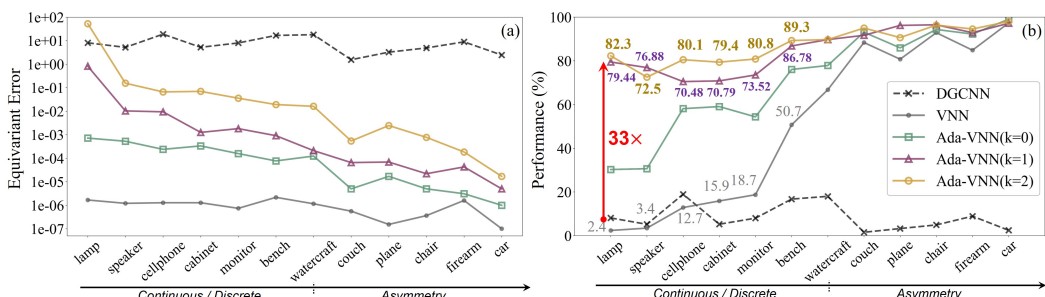

Figure 9: The variation in (a) equivariance and (b) pose estimation performance of Ada-VNN across different categories with varying numbers of DV-MLP hidden layers. The performance metric was Registration Recall (10°). The number of residual layers $l$ is set to 1 in this comparison.

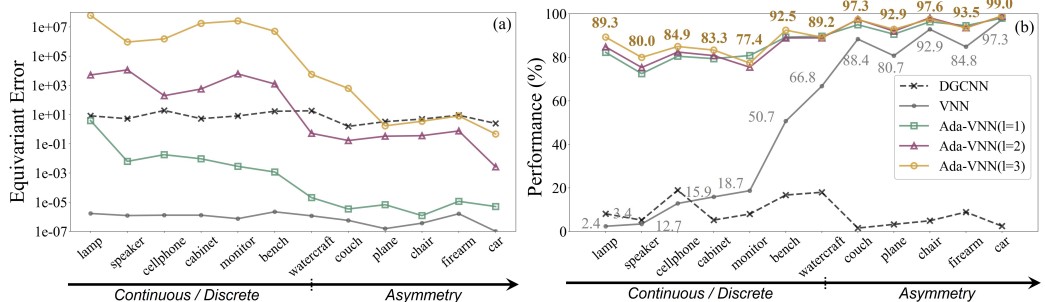

Figure 10: (a) Equivariance performance across different experimental groups. (b) Pose estimation performance across different experimental groups. The performance metric was Registration Recall (10°). The number of hidden layers $k$ in the DV-MLP is set to 1 in this comparison.

| Methods | Memory (GB) | Performance ModelNet40 | Performance ShapeNet |
|---|---|---|---|
| VNN | 10.69 | 10.82 | 35.77 |
| Ada-VNN (l=1) | 10.71 *+0.18%* | 65.78 *+508%* | 60.94 *+70.4%* |
| Ada-VNN (l=2) | 10.75 *+0.37%* | 66.12 *+0.5%* | 62.95 *+3.3%* |
| Ada-VNN (l=3) | 10.80 *+0.46%* | 66.45 *+0.5%* | 63.21 *+0.4%* |
| VN-T | 10.50 | 66.37 | 49.60 |
| Ada-VNT (l=1) | 10.52 *+0.19%* | 82.35 *+24.1%* | 63.55 *+28.1%* |
| Ada-VNT (l=2) | 10.56 *+0.38%* | 84.78 *+3.0%* | 64.23 *+1.1%* |
| Ada-VNT (l=3) | 10.59 *+0.28%* | 85.91 *+1.3%* | 65.51 *+2.0%* |

Table 2: Experiments were conducted with a batch size of 16, where VNN used 1024 feature channels, and VN-T used 512. The average performance metric was 5° Registration Recall (%).

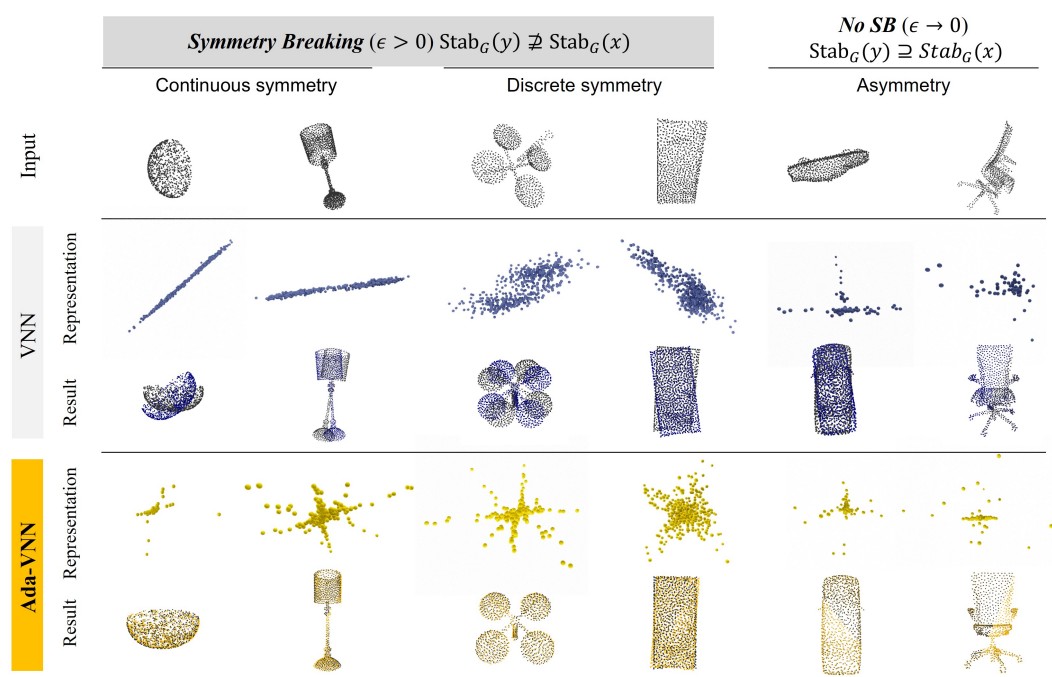

Figure 11: Feature collapse phenomenon and comparative visualization.

