# OpenReview forum: "Towards Adaptive Symmetry Breaking in Vector Neuron Networks"
_ICLR.cc/2026/Conference — ICLR 2026 Conference Withdrawn Submission_

### Official Review · Reviewer_wKHN · 2025-10-20

**Soundness:** 1
**Presentation:** 2
**Contribution:** 2
**Rating:** 2
**Confidence:** 3

**Summary:**

Vector Neuron Networks (VNNs) are a popular form of rotation equivariant neural networks for point cloud processing. The submitted paper studies a symmetry breaking mechanism for VNNs that enables good performance on tasks where the input point cloud might be more symmetric than the expected output. In particular, the problem studied in the experiments is pose regression. When the input point cloud (say, a vase) is symmetric under some rotations then multiple output poses are equally valid. Using a standard VNN works poorly since the network is incapable of breaking the symmetry of the input to output a single pose.

The specific form of symmetry breaking proposed in this paper is a form of Residual Pathway Priors (RPP), where a non-equivariant layer is applied as a residual path in the network. Experiments show that this form of symmetry breaking works well for pose regression of symmetric objects.

Further, the paper presents an analysis of the loss function, claiming that it leads to “a subtle form of adaptive equivariance”.

**Strengths:**

1. The experimental results are strong, effectively demonstrating the improvement over ordinary VNNs in the pose regression task.
2. The experimental results are also well presented, in particular Figure 4 is illustrating the differences over different categories nicely.

**Weaknesses:**

It is unclear to me how exactly the method works, see the Questions below. Furthermore, the discussion in Section 4.2 on “Implicit Equivariance Regularisation” seems inaccurate to me. I will write my concerns here in the hope that they can be addressed.
1. Equation (2) presents an upper bound on the objective function. As it is an upper bound, there is nothing that suggests that (2) would be minimized during training.
2. Indeed, (2) holds equally well with $\rho(g)f_\theta(x)$ in both expectations replaced by $h(x,g)$ where $h$ is any function. This is because the proof of (2) only uses Lipschitz continuity of $D$ and triangle inequality of $d$, i.e. no properties of $G$ or $\rho$.
3. On line 246, $\alpha = \mu(G_s)$ is introduced, where $\mu$ is the Haar measure of $G$ and $G_s$ is a subgroup of $G$ corresponding to the symmetries of some input $x$. We note that all proper subgroups of $SO(3)$ have Haar measure 0. $SO(3)$ is relevant to consider, since the experiments use that group.
4. Thus, for $SO(3)$, equation (5) collapses to two cases, either full symmetry, $\alpha=1$, or less than full symmetry and $\alpha=0$. Additionally, (5) is again an upper bound on the objective, so it is not actually minimized during training.
5. Because Section 4.2 studies upper bounds of the objective, it seems to me like conclusions such as “Therefore, the strength of the equivariance regularization of the $\Psi$ is negatively correlated with the symmetry strength of the input, and the equivariance constraint of Ada-VNNs is adaptively relaxed.” are not correct.

Minor weaknesses/typos:
1. Both $G_s$ and $Stab_G(x)$ used for the subgroup that fixes $x$.

**Questions:**

1. It is unclear to me why the proposed method solves the degeneracy better than ordinary VNNs. Regardless of whether $F$ is equivariant or not, if $g$ fixes $x$, then $F(\rho(g)x)=F(x)$, so informally $F$ will have trouble choosing between regressing $g$ or any other $g’$ that fixes $x$ (since the loss is $d(F(\rho(g)x), g)$). What is it that enables a non-equivariant architecture to handle this better?
3. As stated on line 352, the model takes two point clouds as input and predicts their relative rotation. How is this handled in the VNN-framework? Is it equivariant to the simultaneous rotation of both point clouds, or is it equivariant under independent rotations of the point clouds?

---

> ### Author Response · Authors · 2025-11-22
> **Response to Reviewer wKHN - Part 1**
>
> **Dear Reviewer wKHN:**
>
> We sincerely appreciate your insightful and incisive comments! Your core critique focuses on the mathematical theory (Section 4.2), specifically questioning how our proposed objective function guarantees adaptive equivariance behavior in Ada-VNNs. We are deeply grateful for this observation. Prompted by your feedback, **we have revised the mathematical derivation in Section 4.2, and we can now theoretically prove the adaptive equivariance of Ada-VNNs!** In this response, we first briefly report the revised theoretical proof, followed by a point-by-point response to your specific comments.
>
> **Revised Theoretical Proof.** We aim to demonstrate that optimizing $\mathcal{L}$ forces a reduction in the model's equivariance error $\mathcal{E}(f)$ and enables adaptivity based on the data's intrinsic symmetry.
>
> **Preliminary Definitions**
>
> Equivariance Error (Fei & Deng [1*]): $\mathcal{E}(f) = \mathbb{E} _ {x,g}[\| f(\rho(g) x) - \rho(g) f(x) \|]$.
>
> Symmetry Breaking Assumption (Wang et al. [2*]): The inherent symmetry breaking of the true function $f^\ast$ is bounded by $\epsilon$, i.e., $\sup _ {x,g}\| f^\ast(\rho(g) x) - \rho(g) f^\ast(x) \| = \epsilon$.
>
> Optimization Objective: $\mathcal{L} = \mathbb{E} _ {x,g}[\| f(\rho(g) x) - \rho(g) y \|]$.
>
> **Step 1**: Decomposition of the Equivariance Error.
> Using the triangle inequality of norms, we decompose $\mathcal{E}(f)$:
>
> $$
> \begin{aligned}
> \mathcal{E}(f) &= \mathbb{E} _ {x,g}[\| f(\rho(g) x) - \rho(g) f(x) \|] \\
> &= \mathbb{E} _ {x,g}[\| f(\rho(g) x) - \rho(g)f^\ast(x) + \rho(g)f^\ast(x) - \rho(g) f(x) \|] \\
> &\leq \mathcal{L} + \mathbb{E} _ {x,g}[\| \rho(g)f^\ast(x) - \rho(g)f(x) \|]
> \end{aligned}
> $$
>
> Since $\rho(g) \in \text{SO(3)}$ is a norm-preserving transformation, the second term simplifies to $\mathbb{E} _ {x}[\| f^\ast(x) - f(x) \|]$. Thus:
>
> $$
> \mathcal{E}(f) \leq \mathcal{L} + \underbrace{\mathbb{E} _ {x}[\| f^\ast(x) - f(x) \|]} _ {\text{Residual}}
> $$
>
> **Step 2**: Relationship between residual term and loss function.
> We expand $\mathcal{L}$ using the reverse triangle inequality:
>
> $$
> \mathcal{L} \geq \mathbb{E} _ {x,g} [ \| f(\rho(g) x) - f^\ast(\rho(g)x) \| ] - \underbrace{\mathbb{E} _ {x,g} [ \| f^\ast(\rho(g)x) - \rho(g)f^\ast(x) \| ]} _ {\leq\epsilon}
> $$
>
> For the first term, let $z = \rho(g)x$. We utilize the Volume Invariance of $\rho(g)$ ($|\det(\rho)|=1$) to preserve the integration measure ($dz=dx$). Combined with the standard assumption that the training distribution covers the group orbit (achieved via augmentation), we have the change-of-variables equality: $\mathbb{E} _ {x,g}[\| f(\rho(g)x) - f^\ast(\rho(g)x) \|] = \mathbb{E} _ {x}[\| f(x) - f^\ast(x) \|]$.
>
> Substituting this into the inequality yields:
>
> $$
> \mathbb{E} _ {x}[\| f(x) - f^\ast(x) \|] \leq \mathcal{L} + \epsilon
> $$
>
> **Step 3**: Synthesis.
> Substituting the result from Step 2 into Step 1, we obtain the bound $\mathcal{E}(f) \leq 2\mathcal{L} + \epsilon$.
> Considering the Ada-VNNs structure $f(x) = \Phi(x) + \Psi(\Phi(x))$, since the backbone $\Phi$ is strictly equivariant (satisfying $\Phi(\rho(g) x) = \rho(g) \Phi(x)$), the overall equivariance error is entirely determined by the residual part $\Psi$:
>
> $$
> \mathcal{E}(f) = \mathcal{E}(\Psi) \leq 2\mathcal{L} + \epsilon
> $$
>
> The proof is completed. This inequality demonstrates that by minimizing the proxy loss $\mathcal{L}$, the model's equivariance error is constrained within $2\mathcal{L} + \epsilon$. This reveals an adaptive mechanism:
>
> * When data has no symmetry breaking ($\text{Stab} _ {G}(y) \supseteq \text{Stab} _ {G}(x), \ \epsilon \rightarrow 0$): As $f$ is trained to convergence ($\mathcal{L} \to 0$), $\mathcal{E}(f) \to 0$, and $f$ tends to converge to a strictly equivariant model.
> * When data exhibits symmetry breaking ($\text{Stab} _ {G}(y) \nsupseteq \text{Stab} _ {G}(x),\ \epsilon>0$): As $f$ is trained to convergence ($\mathcal{L} \to 0$), $\mathcal{E}(f) \to \epsilon$, and $f$ tends to converge to an $\epsilon$-weakly equivariant model. This indicates that our optimization objective enables the model $f$ to achieve adaptive equivariance based on the data's symmetry properties.
>
> Notably, the derivation results align perfectly with our experimental findings (Sections 5.1 and 5.2). Due to space limitations, we recommend referring to the revised manuscript for further details on the derivation.
>
> **Below, we will provide a point-by-point response to your comments based on this revised theoretical derivation.**

---

> > ### Author Response · Authors · 2025-11-22
> > **Response to Reviewer wKHN - Part 2**
> >
> > **W1:** Equation (2) presents an upper bound on the objective function. As it is an upper bound, there is nothing that suggests that (2) would be minimized during training.
> >
> > **W5:** Because Section 4.2 studies upper bounds of the objective, it seems to me like conclusions such as “Therefore, the strength of the equivariance regularization of the $\Psi$ is negatively correlated with the symmetry strength of the input, and the equivariance constraint of Ada-VNNs is adaptively relaxed.” are not correct.
> >
> > **Response to W1&5:** Please allow us to address W1 and W5 together, as both critically question the validity of the optimization mechanism in our theoretical derivation. We are very grateful for this point, which is a very sharp and crucial criticism. You are completely correct: in the derivation of the initial draft, we merely showed an upper bound relationship but failed to provide solid mathematical evidence that "minimizing the objective function $\mathcal{L}$ causally leads to the reduction of the equivariance error $\mathcal{E}(f)$," nor did we rigorously prove the existence of the adaptive relaxation mechanism.
> >
> > Deeply inspired by your feedback, we have completely revised the mathematical derivation in Section 4.2. The new derivation no longer relies on vague upper bound assumptions but establishes a strict inequality constraint:
> >
> > $$
> > \mathcal{E}(f) \leq 2\mathcal{L} + \epsilon
> > $$
> >
> > where $\epsilon = \sup \|f^\ast(\rho(g)x) - \rho(g)f^\ast(x)\|$ represents the inherent degree of symmetry breaking in the data distribution.
> >
> > This new conclusion directly addresses your concerns.
> >
> > **Regarding the Minimization Mechanism (to W1):**
> > The new inequality indicates that $\mathcal{L}$ controls the upper bound of $\mathcal{E}(f)$. According to the properties of the inequality, as $\mathcal{L} \to 0$, the model's equivariance error $\mathcal{E}(f)$ is forcibly compressed and must converge within the range of $\epsilon$.
> >
> > **Regarding the Adaptive Conclusion (to W5):**
> > The new formula clearly reveals that the source of the adaptive mechanism is the property of the data itself, $\epsilon$.
> >
> > * When the data has no symmetry breaking ($\epsilon = 0$): The inequality becomes $\mathcal{E}(f) \leq 2\mathcal{L}$. Minimizing $\mathcal{L}$ forces the model to tend towards strict equivariance ($\mathcal{E}(f) \to 0$).
> > * When the data has symmetry breaking ($\epsilon > 0$): The inequality becomes $\mathcal{E}(f) \leq 2\mathcal{L} + \epsilon$. At this time, minimizing $\mathcal{L}$ no longer enforces zero equivariance error but allows a relaxation margin of degree $\epsilon$.
> >
> > This means that the model's equivariance constraint is dynamically determined by the inherent symmetry ($\epsilon$) of the data. This theoretically proves that Ada-VNNs can adaptively adjust their equivariant behavior according to the symmetry strength of the input. This conclusion aligns perfectly with the experimental results observed in Figure 3 of our paper. We have updated Section 4.2 of the revised manuscript with this rigorous proof. Finally, to more intuitively compare the optimization of the objective function with the change in the model's equivariance, we have added curves showing the change of the objective function and the model's equivariance error during training in the revised manuscript to demonstrate the alignment between the experiments and the theory, in Section 5.2.
> >
> > **W2:** Indeed, (2) holds equally well with $\rho(g)f _ \theta(x)$ in both expectations replaced by $h(x,g)$ where $h$ is any function. This is because the proof of (2) only uses Lipschitz continuity of $D$ and triangle inequality of $d$, i.e. no properties of $G$ or $\rho$.
> >
> > **Response to W2:** We sincerely appreciate your insightful feedback and keen mathematical intuition. This reminder led us to leverage these properties to refine our mathematical derivation. In the revised derivation, we utilize two key properties of $\rho$. The first is Norm Preservation: $\| \rho(g) f^\ast(x) - \rho(g) f(x) \| = \| f^\ast(x) - f(x) \|$, which is a crucial step in Step 1 of the derivation. The second is Volume Invariance: $\| f(\rho(g)x) - f^\ast(\rho(g)x) \| = \| f(x) - f^\ast(x) \|$, which forms the basis for Step 2.

---

> > > ### Author Response · Authors · 2025-11-22
> > > **Response to Reviewer wKHN - Part 3**
> > >
> > > **W3:** On line 246, $\alpha = \mu(G _ s)$ is introduced, where $\mu$ is the Haar measure of $G$ and $G _ s$ is a subgroup of $G$ corresponding to the symmetries of some input $x$. We note that all proper subgroups of $\text{SO}(3)$ have Haar measure 0. $\text{SO}(3)$ is relevant to consider, since the experiments use that group.
> > >
> > > **W4:** Thus, for $\text{SO}(3)$, equation (5) collapses to two cases, either full symmetry, $\alpha=1$, or less than full symmetry and $\alpha=0$. Additionally, (5) is again an upper bound on the objective, so it is not actually minimized during training.
> > >
> > > **Response to W3&4:** Please allow us to address Weakness 3 and Weakness 4 together, as both concern the validity of using the Haar measure. This is a very profound and incisive comment, and you are correct. In the initially submitted manuscript, using the Haar measure to quantify the size of subgroups in a continuous group was inappropriate, because in the continuous $\text{SO}(3)$ group, the volume of any symmetry subgroup is 0 unless it is fully symmetric. We hoped $\alpha$ would be a continuous value between $[0, 1]$ to represent the degree of symmetry, but mathematically, it can effectively only take values of $0$ or $1$.
> > >
> > > Inspired by this, we have revised our previous mathematical derivation and made the following improvements in the current mathematical modeling. First, we have discarded $\alpha$ (the Haar measure). Second, we introduced $\epsilon$ (supremum error) from [2*]: $\sup \| f^\ast(\rho(g)x) - \rho(g)f^\ast(x) \| = \epsilon$, which serves as a continuous metric for describing the symmetry breaking of continuous functions. Regardless of whether the volume of $G _ s$ is 0, $\epsilon$ can describe the distance by which $f^\ast$ deviates from strict equivariance. Finally, we established an interpretable inequality: $\mathcal{E}(f) \leq 2\mathcal{L} + \epsilon$. This proves that as long as the Loss ($\mathcal{L}$) is reduced sufficiently low, the model's equivariant error $\mathcal{E}(f)$ must be suppressed to the level of $\epsilon$. Since $\epsilon$ depends on the symmetry of the data itself, the equivariance of model $f$ exhibits a state of alignment with the data symmetry.
> > >
> > > **Minor W1:** Both $G _ s$ and $\text{Stab} _ G(x)$ used for the subgroup that fixes $x$.
> > >
> > > **Response to Minor W1:** We thank the reviewer for the correction. We have unified the notation in the revised manuscript to consistently use $\text{Stab} _ G(x)$.

---

> ### Author Response · Authors · 2025-11-22
> **Response to Reviewer wKHN - Part 4**
>
> **Q2:** As stated on line 352, the model takes two point clouds as input and predicts their relative rotation. How is this handled in the VNN-framework? Is it equivariant to the simultaneous rotation of both point clouds, or is it equivariant under independent rotations of the point clouds?
>
> **Q1:** It is unclear to me why the proposed method solves the degeneracy better than ordinary VNNs. Regardless of whether $F$ is equivariant or not, if $g$ fixes $x$, then $F(\rho(g)x)=F(x)$, so informally $F$ will have trouble choosing between regressing $g$ or any other $g'$ that fixes $x$ (since the loss is $d(F(\rho(g)x), g)$). What is it that enables a non-equivariant architecture to handle this better?
>
> To address your concerns more logically, we first respond to Question 2 to clarify the system architecture, and then build upon that to address Question 1.
>
> **Response to Q2:**
> We follow the setup used in VNN and EPN. Specifically, we employ the proposed encoder $f$ as a feature extractor to process the source point cloud $x$ (canonical pose) and the target point cloud $\rho(g)x$ (transformed pose) respectively, yielding the feature pair $V = \{f(x), f(\rho(g)x)\}$. Subsequently, a simple MLP is used to regress the relative rotation $R = \text{MLP}(V)$ from these features. In this framework, we are concerned with the equivariance of the feature extractor $f$ with respect to the individual input point cloud. For categories without symmetry breaking ($\text{Stab} _ {G}(y) \supseteq \text{Stab} _ {G}(x), \epsilon=0$), the MLP can accurately regress $R$—for instance, by comparing the features via $f(\rho(g)X) f(X)^T$—only if the condition $f(\rho(g) X) \approx \rho(g) f(X)$ holds. Consequently, in this setting, stronger network equivariance is more beneficial for the task.
>
> **Response to Q1:**
> This question is critical, as it touches upon the core mechanism underlying the effectiveness of our method.
>
> First, we fully agree with your view: if there is perfect mathematical symmetry (e.g., a perfect sphere), no deterministic function can distinguish $x$ from $\rho(g)x$. However, VNN deals with point cloud data. Real-world object point clouds (like Can, Lamp categories) usually exhibit **approximate symmetry**. They contain high-frequency geometric details that break perfect symmetry, such as the pull-tab on a soda can, side buttons on a phone, surface textures, or even random noise from point cloud sampling.
>
> Strictly equivariant networks enforce that the output transforms strictly with the input. This strong constraint leads to the collapse of equivariant features. Assume input $x$ is approximately symmetric with respect to a group $G_s$, meaning for any group element $g_s \in G_s$, we have $\rho(g_s)x \approx x$. We define the deviation as $\delta = \|\rho(g_s)x - x\|$. For a strictly equivariant function $f$, the condition $f(\rho(g_s)x) = \rho(g_s)f(x)$ must hold. Using Lipschitz continuity (constant $k$), we have:
>
> $$
> \|\rho(g_s)f(x) - f(x)\| = \|f(\rho(g_s)x) - f(x)\| \leq k \|\rho(g_s)x - x\| = k \delta
> $$
>
> The term on the left, $\|\rho(g_s)f(x) - f(x)\| \leq k\delta$, implies that when the input is very close to symmetric ($\delta \to 0$), the output feature $f(x)$ is forced to approach the Fixed Point Set of the group $G_s$.
>
> If $G_s$ represents a rotational symmetry group about a specific axis, the projection of any vector $f$ satisfying $\rho f \approx f$ onto the space orthogonal to this axis must be a very short vector (approaching 0). This phenomenon is what we term **feature collapse** when strictly equivariant models handle symmetry breaking. This feature collapse suppresses the critical geometric information needed to distinguish orientation, causing the regression layer to fail in predicting the rotation.
>
> Relaxed equivariant functions are not bound by the strict equality above. This means that even if $\delta$ is small, the relaxed equivariant function is allowed to produce features where $\rho(g_s)f(x) \neq f(\rho(g_s)x)$. This flexibility enables the model to amplify and exploit those tiny "asymmetries" to resolve ambiguity. Informally speaking, strictly equivariant networks treat "asymmetry" as noise to be filtered out, while relaxed equivariant networks amplify these signals for utilization. This further highlights the advantage of our method: since the symmetry in data is usually unknown, fixed relaxed equivariance is often sub-optimal. Our method can dynamically adjust the degree of relaxation according to the actual symmetry of the data. We added Section 5.4 ("Why Ada-VNNs Work") to the revision to analyze why our method outperforms VNN. Furthermore, we provided comparative feature maps of VNN and Ada-VNN to visually demonstrate the representation collapse phenomenon (Figure 5).

---

> > ### Author Response · Authors · 2025-11-22
> > **Response to Reviewer wKHN - Part 5**
> >
> > **Closing Remarks**
> >
> > Once again, we extend our sincere gratitude for your constructive criticism, which was instrumental in rectifying the theoretical framework of our work. We believe the revised manuscript now offers a mathematically rigorous and empirically consistent explanation for the adaptive equivariance of Ada-VNNs. We hope these comprehensive revisions and detailed responses satisfactorily address your concerns.
> >
> > Finally, we deeply appreciate the time you spent reviewing our manuscript and the high quality of your feedback. Thank you for your contribution to the open AI community. As fellow researchers, we also wish you the best of luck with your own rebuttals during this season. We look forward to any further discussion during the interaction period and remain fully committed to addressing any remaining questions.
> >
> > Sincerely, The Authors
> >
> > **References**
> > [1] Fei, Jiajun, and Zhidong Deng. "Rotation invariance and equivariance in 3D deep learning: a survey." Artificial Intelligence Review 57.7 (2024): 168.
> >
> > [2] Wang, Rui, Robin Walters, and Rose Yu. "Approximately equivariant networks for imperfectly symmetric dynamics." International Conference on Machine Learning. PMLR, 2022.

---

> > > ### Comment · Reviewer_wKHN · 2025-11-28
> > >
> > > I thank the authors for their response and am happy to see the theory revised. I will say that it should not be expected from reviewers to review an entirely new theoretical contribution replacing a key section of the paper. From this perspective I would recommend the authors to submit this new version to another venue.
> > >
> > > Nevertheless, I have had a look at the new theory and find it much more adequate than the previous version. I have two remaining concerns.
> > >
> > > 1. I think the new discussion in Section 4.2 is accurate. However, I think it is closely aligned with the discussion by Wang et al. (2022). For instance, I think the connection to Wang et al.'s Proposition 3.3 should be covered in depth.
> > > 2. The new discussion in Section 5.4 is quite interesting. It seems to me like the issue with using an equivariant VNN is that the final pose regressing MLP is not capable of doing symmetry breaking from the output of the VNN. This seems plausible. What I still do not understand is how the loss works for the Ada-VNN mode.l I still do not see how the Ada-VNN can "decide" between outputting different poses in the stabilizer if the training loss is on a specific pose. For instance, for the SO(2) symmetric lamp in Figure 5, are you saying that there is exploitable info in the noise in the point cloud that allows symmetry breaking? If so, is the performance of the model worse if during training we add SO(2) augmentations to the lamp (and not to the target pose) so that this noise is inconsistent between training epochs?

---

> > > > ### Author Response · Authors · 2025-11-28
> > > > **Response to Reviewer wKHN - Stage2  Part1**
> > > >
> > > > **Dear wKHN:**
> > > >
> > > > Thank you for your response. We are pleased that we have addressed your previous concerns. Below, we provide detailed responses to your two remaining points.
> > > >
> > > >
> > > >
> > > > **A1:** We appreciate the reviewer’s validation of our revised Section 4.2. While we indeed adopt the assumption from Wang et al. (2022) regarding intrinsic symmetry breaking (i.e., $\sup || f^{\ast}(\rho(g) x)-\rho(g)f^{\ast}(x) ||=\epsilon$) to ensure rigor, our analysis and conclusions go a step further than their Proposition 3.3.
> > > >
> > > > The key distinctions are as follows:
> > > >
> > > > 1.  Proposition 3.3 (Wang et al. ) assumes that if a model $f$ approximates the target $f^{*}$ well (i.e., the approximation error $c$ is small), then the equivariance error is bounded. It describes a static property of the function space ("If this state is reached, equivariance is guaranteed") but does not explain how the optimization process reaches this state via the loss function.
> > > >     Our Work (Section 4.2). We derive an optimization-centric bound: $\mathcal{E}(f) \leq 2\mathcal{L}+\epsilon$. This directly links the task loss $\mathcal{L}$ to the equivariance error. It proves the mechanism of adaptive equivariance: as long as the model performs well on the prediction task (minimizing $\mathcal{L}$), the equivariance error $\mathcal{E}(f)$ is forced to decrease (bounded by the intrinsic margin $\epsilon$).
> > > > 2.  Our theoretical result explains why Ada-VNN achieves adaptive equivariance spontaneously during training without requiring explicit regularization terms, as exemplified in pose estimation tasks. In contrast, Wang et al. typically rely on explicit equivariance regularization (e.g., soft constraints on weights $\|w_i(h) - w_i(g)\|$) to achieve their results. Our theory demonstrates that for our task, the task loss itself acts as an implicit regularizer.
> > > >
> > > > Beyond the specific proof in Section 4.2, our work differs substantially from Wang et al. (2022) in overall contribution and scope. As detailed in our response to Reviewer kASe (Part 7), we highlight five key distinctions:
> > > >
> > > > 1. **Distinct Mechanisms for Symmetry Breaking.** Wang et al. (2022) achieve this by relaxing the weight-sharing constraint of convolution kernels  (e.g., making kernel weights dependent on group element pairs $(g, h)$ rather than just $g^{-1}h$). In contrast, Ada-VNNs introduce a residual architecture, where a symmetry-breaking lightweight branch $\Psi$ is appended to a strictly equivariant backbone $\Phi$ ($f(x) = \Phi(x) + \Psi(\Phi(x))$). Furthermore, we design the Decoupled Vector MLP (DV-MLP), which employs three independent weight matrices ($W^1, W^2, W^3$). Symmetry breaking is controlled by the divergence among these matrices, a mechanism specific to  Ada-VNNs.
> > > > 2. **Distinct Mechanisms for Adaptivity (Implicit vs. Explicit).** Wang et al. (2022) rely on explicit regularization, requiring an additional penalty term in the loss function (e.g., $||w_i(h) - w_i(g)||$) and manual tuning of hyper-parameters. Ada-VNNs utilize implicit regularization: we theoretically prove that no additional equivariance penalty is needed. By simply minimizing the task prediction loss $\mathcal{L}$, the model's equivariance error $\mathcal{E}(f)$ is automatically bounded by the data's intrinsic symmetry breaking $\epsilon$.
> > > > 3. **Different Theoretical Focus.** Wang et al. (2022) primarily prove that strictly equivariant models cannot approximate approximately equivariant ground truths. Building upon this, Ada-VNNs go further to derive an adaptive mechanism directly from the objective function. Additionally, we provide a theoretical analysis of  feature collapse, explaining why strictly equivariant models fail in symmetry-breaking tasks
> > > > 4. **Different Base Architectures and Groups.** Wang et al. (2022) are based on Convolutional Neural Networks (Group/Steerable CNNs), focusing primarily on 2D planar symmetry groups (e.g., $SO(2)$). Ada-VNNs are built upon the Vector Neuron framework, specifically addressing the 3D rotation group $SO(3)$.
> > > > 5. **Different Application Domains and Data Objects.** Wang et al. (2022) primarily process 2D vector fields or grid-based data (e.g., smoke simulations, turbulent velocity fields, and ocean currents) , whereas Ada-VNNs mainly handle 3D data (e.g., point clouds).

---

> > > > ### Author Response · Authors · 2025-11-28
> > > > **Response to Reviewer wKHN - Stage2  Part2**
> > > >
> > > > **A2:** Thank you for this insightful question; we would like to offer the following clarification. Ada-VNNs are not required to strictly satisfy the equivariance constraint (i.e., $f(\rho(g _ s)x) \neq \rho(g _ s)f(x)$), which effectively alleviates the feature collapse issue inherent to standard VNNs. Driven by minimizing the objective $\mathcal{L}$, Ada-VNNs learn to leverage minute input asymmetries $\delta$ to form directionally distinct representations, enabling the output of different poses. The lamp in Figure 5 represents an extremely challenging case, where asymmetry arises primarily from non-uniform sampling and noise. However, most objects in the Lamp category—and real-world objects in general—possess distinct local structural asymmetries, such as buttons or pull cords. Regarding your specific query: during training, random rotations are applied to the template point cloud, and sampling is performed independently for each instance. Consequently, even if we add SO(2) augmentation to the lamp so that noise patterns are inconsistent across epochs, the model's performance does not degrade. We will further refine Section 5.4 to prevent any ambiguity.
> > > >
> > > >
> > > >
> > > >
> > > >
> > > > **Response regarding the recommendation to submit to another venue:**
> > > >
> > > > We thank the reviewer for the interest in the discussion in Section 5.4 and the acknowledgment of the quality of the revised Section 4.2. Regarding the recommendation to submit this new version to another venue, we respectfully disagree.   We wish to clarify that this revision does not constitute an entirely new theoretical contribution, but rather a correction and strengthening of the original theoretical analysis.    Our reasoning is as follows:
> > > >
> > > > **1.      Consistency of Core Contributions:**
> > > > The core contributions of our paper—the Ada-VNN architecture and the adaptive equivariance mechanism—remain exactly the same as in the original submission.      The contributions listed in our Introduction have not changed.      The revised Section 4.2 still serves to support the **same conclusion**: that minimizing the task loss $\mathcal{L}$ implicitly controls the equivariance error.     The revised Section 4.2 merely adopts a more robust mathematical tool (the assumption from Wang et al., 2022) to fix the technical gaps in the original proof.     This is a correction and strengthening of the derivation for an existing claim, not a change in the paper's fundamental contribution.
> > > >
> > > > **2.      The Purpose of the Rebuttal Process:**
> > > > The core value of peer review and the rebuttal process is precisely to eliminate errors and improve paper quality through feedback.      Since you also acknowledge that the new section is "interesting" and more plausible, this improvement is a testament to the success of the current review process.      Rejecting a submission because the authors actively responded to feedback to substantially improve rigor would seem contrary to the spirit of ICLR, which encourages authors to perfect their work.
> > > >
> > > >
> > > >
> > > >
> > > >
> > > > **Closing Remarks**
> > > >
> > > > Finally, we would like to express our sincere gratitude for your time and effort. We are fully aware that reviewing is a voluntary service to the community. As fellow authors currently navigating the rebuttal process ourselves, we deeply understand the challenges of balancing high-quality reviewing with the workload of one's own submissions, especially given the increasing volume of papers. Therefore, we truly value your constructive feedback and your acknowledgment of the improvements in our revised manuscript. We also wish you the very best of luck with your own submissions during this period.Given that the revised version has significantly strengthened the theoretical rigor and overall quality of the paper, we respectfully request that you re-evaluate our contribution based on these improvements. We would be deeply grateful if you would consider raising your score.
> > > >
> > > > We thank you once again for your dedicated efforts and valuable feedback throughout the review process!
> > > >
> > > > Sincerely, The Authors

---

### Official Review · Reviewer_9Nai · 2025-10-29

**Soundness:** 3
**Presentation:** 3
**Contribution:** 3
**Rating:** 6
**Confidence:** 4

**Summary:**

The paper presents an adaptive framework that unifies equivariance and (different levels of) symmetry breaking into the Vector Neuron networks. The network is designed through a high-level idea of y(x) = equi_net(x) + residual(equi_net(x)). Based on this, the paper proposes some theoretical justifications and network designs. The adjustment to the network architecture is lightweight and easy to understand. Experiments show the network's different levels of equivariance on different object categories with different symmetries.

**Strengths:**

- The paper studies an important problem in equivariant learning, and it presents an interesting and inspiring idea on equivariant network symmetry breaking, with theoretical analysis and explanations.
- The paper proposes a compact network design that unifies different levels of equivariance/symmetry breaking. The adjustments are lightweight.
- The experimental results are interesting and well-supportive of the different levels of symmetry breaking in the network, especially Figure 3 and 4.

**Weaknesses:**

- It seems that to compute the objective with symmetry breaking (to my understanding, Eq. 5, or Eq.7, is the loss function to train the network), one needs to know what the symmetry group is ahead of time, in order to compute $\alpha$. The (category-level) pose estimation task is a good justification for this setting, but I feel that for more general applications, knowing the symmetry group (for each object) ahead of time is less practical.
- How could the symmetry-breaking objective be applied to discrete symmetries? The finite subgroups (for discrete symmetries) are of measure zero in SO(3)?
- The writing of the paper can be made clearer, especially for some technical details that are crucial to the method. See Questions for more.

**Questions:**

- Is Eq. 5 the ultimate loss function used to train the network?
- Why is it called an "upper bound"? -- I think even if a loss cannot be minimized to zero, which is most common in deep learning, it can just be called the "objective" and no need to call it an "upper bound". When saying the "upper bound", I would wonder what is the actual loss function is to be optimized.

---

> ### Author Response · Authors · 2025-11-22
> **Response to Reviewer 9Nai - Part 1**
>
> **Dear Reviewer 9Nai:**
>
> We sincerely appreciate your valuable comments!  Your concerns primarily focus on two aspects: first, clarifying the specific objective function of Ada-VNNs (Q1, Q2);  and second, understanding how this objective achieves adaptive equivariance, such as determining $\alpha$ (W1) and its applicability to discrete symmetries (W2). We acknowledge that the description of these mechanisms in our initial manuscript was not sufficiently clear.  Motivated by your feedback, we have extensively revised the Methodology section (specifically Section 4.2) and re-formulated the optimization problem.  Based on this refined framework, we are now able to fully address your concerns. Please allow us to respond to your questions in a reorganized order below.
>
> **Question1:** Is Eq. 5 the ultimate loss function used to train the network?
>
> **A1:** Eq. 5 is not our training objective; rather, it is an inequality derived to quantify the relationship between the optimization objective and the model's equivariance error. The actual objective function we employ is the general formulation for equivariant tasks:
> $$\mathcal{L} = \mathbb{E}_{x,g}[\| f(\rho(g) x) - \rho(g) y \|]$$
>
> It is important to clarify that we are not proposing a "new" objective function in the traditional sense. Instead, we have derived a novel adaptive equivariance mechanism from the existing objective to enable flexible symmetry breaking. For instance, in SO(3) tasks where the model predicts the rotation applied to an input, previous approaches often relied on strictly equivariant networks (e.g., VNNs). However, real-world data distributions often exhibit spontaneous and unknown symmetry breaking (approximate equivariance). A classic analogy is the "Mexican Hat" potential: while the potential function itself is rotationally symmetric, the system must settle into a specific low-energy stable point as the output.
>
> In such scenarios, strictly equivariant functions are inapplicable because they cannot infer lower-symmetry outputs from higher-symmetry inputs. Consequently, we derived an adaptive mechanism from $\mathcal{L}$. This theoretically demonstrates that by minimizing this objective, a model can acquire equivariance that naturally adapts to the data's intrinsic symmetry—without requiring a symmetry prior. Our Ada-VNN framework effectively leverages this mechanism to achieve adaptive equivariance.
>
> We have revised Section 4.2 to define our objective function more clearly.
>
>
> **Question2:** Why is it called an 'upper bound'? -- I think even if a loss cannot be minimized to zero, which is most common in deep learning, it can just be called the "objective" and no need to call it an "upper bound". When saying the "upper bound", I would wonder what is the actual loss function is to be optimized.
>
> **A2:**
> In Section 4.2, we introduced the concept of an 'upper bound' to derive the relationship between the optimization objective and the model's equivariance error. Our goal is to demonstrate: 'Whether optimizing this objective enables Ada-VNNs to achieve adaptive equivariance across data with varying symmetries.' Please allow us to clarify this issue using the revised derivation
>
> First, for Ada-VNN, the quantified Equivariance Error is defined as:
> $$
> \mathcal{E}(f) = \mathbb{E}_{x,g}[\| f(\rho(g) x) - \rho(g) f(x) \|]
> $$
>
> The Optimization Objective is:
> $$
> \mathcal{L} = \mathbb{E}_{x,g}[\| f(\rho(g) x) - \rho(g) y \|]
> $$
>
> We assume the inherent symmetry breaking of the ground-truth physical process $f^*$ is bounded by $\epsilon$, i.e.,
> $$
> \sup_{x,g}\| f^*(\rho(g) x) - \rho(g) f^*(x) \| = \epsilon
> $$
> (This measures the intrinsic degree of symmetry breaking, which is generally unknown).
>
> Our derived conclusion is:
> $$
> \mathcal{E}(f) \leq 2\mathcal{L} + \epsilon
> $$
> This indicates that the term $2\mathcal{L} + \epsilon$ constitutes an upper bound on the model's equivariance error $\mathcal{E}(f)$. We can control the model's equivariance error by regulating $\mathcal{L}$ and $\epsilon$.
>
>
> Specifically:
> * **No Symmetry Breaking** ($\mathrm{Stab}_G(y) \supseteq \mathrm{Stab}_G(x)$, $\epsilon \to 0$): As $f$ is trained to convergence ($\mathcal{L} \to 0$), $\mathcal{E}(f) \to 0$, meaning $f$ tends to converge to a strictly equivariant model.
> * **Symmetry Breaking** ($\mathrm{Stab}_G(y) \nsupseteq \mathrm{Stab}_G(x)$, $\epsilon > 0$): As $f$ is trained to convergence ($\mathcal{L} \to 0$), $\mathcal{E}(f) \to \epsilon$, meaning $f$ tends to converge to an $\epsilon$-weakly equivariant model.
>
> This indicates that Ada-VNNs are capable of achieving adaptive equivariance based on the data's symmetry properties.
>
> Continued in Part 2...

---

> > ### Author Response · Authors · 2025-11-22
> > **Response to Reviewer 9Nai - Part 2**
> >
> > **Weakness2:** How could the symmetry-breaking objective be applied to discrete symmetries? The finite subgroups (for discrete symmetries) are of measure zero in SO(3)?
> >
> > **Response to Weakness2:** We sincerely appreciate the reviewer's keen observation regarding this issue. We confirm that for a continuous group like $\mathrm{SO(3)}$, any finite subgroup corresponding to discrete symmetries (e.g., $C_n$) indeed has zero Haar measure under the continuous measure. This validates the criticism that our initial derivation, which relied on the continuous Haar measure (e.g., the $\alpha$ parameter), was flawed for discrete symmetries.
> >
> > To address this issue, we have introduced a theoretical framework based on the quantitative supremum error $\epsilon$: specifically, we utilize Assumption 1 ($\sup_{x,g} \| f^\ast(\rho(g) x) - \rho(g) f^\ast(x) \| = \epsilon$) to replace the original measure-theoretic definition. $\epsilon$ is a supremum bound that is explicitly defined for both continuous and finite discrete groups.
> >
> > Consequently, when applying our objective function to discrete symmetries, the expectation operator $\mathbb{E}_{x,g}$ naturally reduces to a uniform summation over the finite set of discrete group elements. Since the core mechanism relies on bounding the equivariance error ($\mathcal{E}(f)$) by the loss ($\mathcal{L}$) and the intrinsic deviation ($\epsilon$), the inequality $\mathcal{E}(f) \leq 2\mathcal{L} + \epsilon$ remains mathematically sound and robustly applies across all types of symmetry.
> >
> > **Weakness1:** It seems that to compute the objective with symmetry breaking (to my understanding, Eq. 5, or Eq.7, is the loss function to train the network), one needs to know what the symmetry group is ahead of time, in order to compute $\alpha$. The (category-level) pose estimation task is a good justification for this setting, but I feel that for more general applications, knowing the symmetry group (for each object) ahead of time is less practical.
> >
> > **Response to Weakness1:** Your question is crucial for understanding our method, as it reflects the key idea of our approach. The goal of our method is to achieve adaptive symmetry breaking without requiring prior knowledge of the data's symmetry prior. We need to clarify that Equation 5 or 7 was not our loss function, but rather a part of the theoretical derivation. We have updated the latest derivation and look forward to your review of the newly submitted version.
> >
> >
> > **Weakness3:** The writing of the paper can be made clearer, especially for some technical details that are crucial to the method.
> >
> > **Response to Weakness3:** We sincerely appreciate your constructive feedback regarding the presentation of our paper. We recognize that clarity is paramount for conveying technical details effectively. In the revised manuscript, we have made significant efforts to enhance readability and precision. First, We have completely rewritten Section 4.2 to provide a rigorous, step-by-step derivation of the implicit regularization bound, replacing the previous measure-theoretic definition with the robust supremum error $\epsilon$. Second, We added a dedicated discussion section ('Why Ada-VNNs Work?', Sec. 5.4) along with new visualizations (Fig. 5) to intuitively explain the mechanism of our adaptive approach.
> >
> >
> > **Closing Remarks:**
> > We would like to express our sincere gratitude to the reviewer for the time and effort dedicated to reviewing our paper. We are particularly grateful for the comments regarding the mathematical formulation (Weakness 2 and 3), which prompted us to clarify our method and develop a much more rigorous theoretical framework based on $\epsilon$-approximate equivariance. We hope that our responses and the revised manuscript have addressed your concerns. Regarding the clarity of the writing, we are committed to making continuous improvements until the rebuttal process is fully concluded. Finally, we look forward to further discussion during the interaction period and remain fully dedicated to addressing any remaining questions you may have.
> >
> > Sincerely,
> >
> > The Authors

---

### Official Review · Reviewer_BbhT · 2025-10-31

**Soundness:** 3
**Presentation:** 2
**Contribution:** 3
**Rating:** 8
**Confidence:** 2

**Summary:**

the paper proposes a variant of the vector neuron network called Adaptive-VNN. It's core contribution is an architectural modification to relax the strict SO3 equivariance assumption when dealing with symmetry breaking. The authors motivate the work raising an issue that strict equivariance constraints prevent networks from inferring low-symmetry outputs like a precise pose from high-symmetry inputs like a can, and that the larger the object symmetry, the worse the performance. the suggested residual architecture maps equivariance constraints as priors, and the level of actual equivariance is dynamically tuned to the input. Experiments on 3D pose estimation show dramatic performance gains compared to standard VNN.

**Strengths:**

+ the paper tackles a limitation of current fully equivariant networks. It analyses the limitation from a theoretical standpoint and also demonstrates this through experimental evidence.
+ the suggested solution is simple conceptually, and immediate to implement. It shows convincing perfroamce gains

**Weaknesses:**

(-) The motivating example (coca cola can) isn't clear to me. Stepping back for a second, the SO3 equivariance isn't about the object itself being symmetric it's about having the predcitions rotate with the input. So the way I understand the problem is that under strict equivariance, given a symmetry transform t that keeps the input sample x in tact, we'd get f(x) = f(t*x) = tf(x), namely the features of a equivairant network become *invariant* to the symmetry. This could hurt tasks that require asymmetry tasks like telling left from right or regressign the exact relative pose. But here it seems the author emphasize a nuanced issue where that the network cannot identify symmetry breakin cues like the can opening direction, since the details are minor. I'm failing to see a principled explanation of why this would be the case -- is this a matter of capacity? I suggest the authors discuss this sensitivity of VNN

(-) the method is shown with VNN and the authors do explain the convenience of having VN maintaining the dimensionality allowing for simple residual connections however seeing this with other architectures would highly push the generality of the idea. I would especially like to see it demonstrated with the frame averaging framework which to my understanding should also allow a residual ingredient pretty much out of the box.

(-) For perfectly symmetric objects, the task of pose estimation also becomes ill posed as many solutions are equally viable for registration. Thus measuring angular error seems problematic -- this changes of course once the symmetry is violated (like in the coca cola can example) but I didn't see a discussion of this symmetry breaking in the experimental design / datasets.

minor:
(-) im missing a description of the role of the decoder in the pose estimation task
(-) 041: "​​the underlying physics" is very vague. would be better if tha autrhos spell out what they mean by that

missing references:
Robust Symmetry Detection via Riemannian Langevin Dynamics
Approximately Piecewise E(3) Equivariant Point Networks
Frame Averaging for Equivariant Shape Space Learning
Equivariant Frames and the Impossibility of Continuous Canonicalization

**Questions:**

see weaknesses

---

> ### Author Response · Authors · 2025-11-24
> **Response to Reviewer BbhT - Part 1**
>
> **Dear Reviewer BbhT:**
>
> We sincerely appreciate your insightful comments and the important questions you have raised. Your concerns primarily focus on four key aspects: (1) a detailed explanation and discussion regarding minute symmetry breaking (e.g., the Coca-Cola example); (2) the missing description of the decoder's role in the experiments; (3) the ambiguity of the term "the underlying physics"; and (4) the citation of relevant prior works. These comments have been instrumental in enhancing the quality of our manuscript. We are confident that we can fully address your concerns. Furthermore, we have rigorously revised the manuscript based on your feedback.
>
>
>
> **W1:** The motivating example (coca cola can) isn't clear to me. Stepping back for a second, the SO3 equivariance isn't about the object itself being symmetric it's about having the predcitions rotate with the input. So the way I understand the problem is that under strict equivariance, given a symmetry transform t that keeps the input sample x in tact, we'd get f(x) = f(t*x) = tf(x), namely the features of a equivairant network become *invariant* to the symmetry. This could hurt tasks that require asymmetry tasks like telling left from right or regressign the exact relative pose. But here it seems the author emphasize a nuanced issue where that the network cannot identify symmetry breakin cues like the can opening direction, since the details are minor. I'm failing to see a principled explanation of why this would be the case -- is this a matter of capacity? I suggest the authors discuss this sensitivity of VNN
>
> **W3:** For perfectly symmetric objects, the task of pose estimation also becomes ill posed as many solutions are equally viable for registration. Thus measuring angular error seems problematic -- this changes of course once the symmetry is violated (like in the coca cola can example) but I didn't see a discussion of this symmetry breaking in the experimental design / datasets.
>
> **Response to W1&W3:** Please allow us to address these two comments together, as they touch upon the core mechanism underlying the effectiveness of our method.
>
> First, we fully agree with your premise: for objects with **perfect mathematical symmetry** (e.g., a perfect sphere), the pose estimation task is indeed ill-posed, as $x$ and $\rho(g)x$ are geometrically indistinguishable, and no deterministic function can recover a unique rotation. However, real-world data (especially point cloud data processed by VNNs) is rarely perfect. Real objects (such as the Can and Lamp categories in ShapeNet) typically exhibit **Approximate Symmetry**. They contain high-frequency geometric details that break perfect symmetry—such as the pull-tab on a soda can, side buttons on a phone, surface textures, or even random noise from point cloud sampling. These are precisely the "asymmetry cues" we focus on.
>
> Regarding the "principled explanation" you requested in W1 regarding VNN's sensitivity, we propose the following derivation:
>
> Assume an input $x$ is approximately symmetric with respect to a group $G_s$, meaning for any $g_s \in G_s$, $\rho(g_s)x \approx x$. We define the deviation as $\delta = \|\rho(g_s)x - x\|$. The closer the object is to perfect symmetry, the smaller $\delta$ becomes.
>
> For a strictly equivariant function $f$, the condition $f(\rho(g_s)x) = \rho(g_s)f(x)$ must hold. Leveraging Lipschitz continuity (with constant $k$), we have:
>
> $$\|\rho(g_s)f(x) - f(x)\| = \|f(\rho(g_s)x) - f(x)\| \leq k \|\rho(g_s)x - x\| = k \delta$$
>
> This implies that as the input approaches perfect symmetry ($\delta \to 0$), the output feature $f(x)$ of the strictly equivariant function is forced to satisfy $\rho(g_s)f(x) \approx f(x)$.
>
> This leads to **Feature Collapse**: If the group $G_s$ represents continuous rotational symmetry about an axis, the only vectors satisfying the condition (fixed points) are those aligned with the rotation axis. Consequently, the components of $f(x)$ orthogonal to the rotation axis—which carry the critical geometric information for distinguishing orientation—are forced to vanish (magnitude $\to 0$). This explains why strictly equivariant networks fail when handling minute asymmetries: it is not a matter of insufficient model capacity, but rather that the mathematical constraints are too rigid to amplify these minute directional signals.
>
> In contrast, Ada-VNNs are not bound by this strict equality. Even when $\delta$ is small, the model is allowed to learn features where $\rho(g_s)f(x) \neq f(\rho(g_s)x)$. This flexibility enables the model to amplify those minute "asymmetries" as signals to resolve ambiguity.
>
> In response to W1 and W3, We have **added a new Section 5.4 (Why Ada-VNNs Work?)** in the revised manuscript, which discusses this mechanism in detail and provides visualizations of the feature collapse phenomenon in VNNs.

---

> ### Author Response · Authors · 2025-11-24
> **Response to Reviewer BbhT - Part 2**
>
> **W2：**:the method is shown with VNN and the authors do explain the convenience of having VN maintaining the dimensionality allowing for simple residual connections however seeing this with other architectures would highly push the generality of the idea. I would especially like to see it demonstrated with the frame averaging framework which to my understanding should also allow a residual ingredient pretty much out of the box.
>
> **Response to W2:**
>
> We thank you for providing such an insightful research direction!  The Frame Averaging (FA) approach is an impressive equivariant method for handling piecewise Euclidean transformation tasks.  Specifically, its property of maintaining constant dimensionality in both the encoder and decoder naturally supports residual connections.  Therefore, our Adaptive Residual Path mechanism can, in theory, be elegantly applied to the FA framework.
>
> Regarding a demonstration during the rebuttal period: Completing this demonstration requires substantial time for investigating specific mathematical details, configuring the development environment, training the model, and reporting the results.  Frame Averaging is an excellent piece of work, and this integration represents a significant learning opportunity for us.  However, we cannot guarantee completion of this comprehensive demonstration within the current rebuttal window.
>
> We assure you of our commitment to this idea: If we are unable to finish the demonstration during the rebuttal period, we will report the subsequent results via the provided open-source code link.  Finally, we will also add the idea of integrating the FA framework into the  section 6 (**"Future Work"**) of the revised manuscript.
>
>
>
> **minor W1:** im missing a description of the role of the decoder in the pose estimation task
>
> **Response to minor W1:** Regarding the role of the decoder in the pose estimation task, we strictly followed the experimental settings established in EPN and E2PN.  Specifically, the proposed model functions as a feature extractor to derive representations from the input point cloud, followed by a simple MLP acting as a decoder to directly regress rotation parameters.
>
> The rationale behind this setup is to assess whether the model's representations are beneficial for the downstream pose estimation task by comparing performance.  This approach is analogous to the linear probing protocol, aiming to provide a fair comparison of representation quality derived from different backbones.  We have **added this detailed explanation to Appendix 9.3 (EXPERIMENTAL DETAILS)** of the revised manuscript.
>
>
>
> **minor W2:** 041: "the underlying physics" is very vague. would be better if tha autrhos spell out what they mean by that
>
> **Response to minor W2:** We appreciate the reviewer for pointing out this ambiguity.  By "underlying physics," we originally intended to describe the intrinsic laws or mappings of the task itself, decoupled from coordinate transformations (e.g., rotation).
>
> For example, in a robotic grasping task, the relationship between the gripper's orientation and a mug's handle is equivariant. If an equivariant network is used, the model can focus solely on learning the grasping strategy (the intrinsic task mapping), rather than wasting capacity learning the geometric transformation rules (i.e., symmetry) from massive data augmentation.
>
> We agree that "physics" might be misleading in a general context. We have revised the sentence to: "...whereas VNNs embed them by network structure design, allowing the model to focus directly on learning the **intrinsic task mapping**."
>
>
>
>
>
> **missing references**
>
>
> We appreciate the reviewer for pointing out these missing references. We agree that these works are strongly related to our study and have now cited them in the corresponding sections of the **Introduction** and **Related Works**.
>
>
>
> **Closing Remarks**
>
> We once again extend our sincere gratitude for your comments. We hope these comprehensive revisions and detailed responses satisfactorily address your concerns. As fellow researchers, we also wish you the best of luck with your own rebuttals this season. Finally, we deeply appreciate the time you dedicated to reviewing our manuscript and your contribution to the open AI community. We look forward to any further discussion during the interaction period and remain fully committed to resolving any remaining issues.
>
> Sincerely, The Authors

---

### Official Review · Reviewer_kASe · 2025-11-03

**Soundness:** 2
**Presentation:** 2
**Contribution:** 2
**Rating:** 2
**Confidence:** 3

**Summary:**

This paper proposes a method to break input symmetry in the context of the Vector Neurons architecture based on residual path priors. A linear non-equivariant layer is added through a residual path to the output of the equivariant vector neuron layer. The paper explains the potential benefits of this approach through an adaptive symmetry breaking effect. Experiments are conducted on pose estimation and confirm this effect on one dataset.

**Strengths:**

- The paper tackles the interesting problem of breaking input symmetries in equivariant learning, following a recent line of work
- The results of the experimental analysis provide interesting insights
- The figures are clear and insightful

**Weaknesses:**

- I think the contributions of this paper are relatively incremental with respect to the existing works on relaxing equivariance works, especially Wang et al. 2022. The method proposed in this paper is essentially an adaptation of Wang et al 2022 to the Vector Neuron architecture.
- Definition 4 is inconsistent with the accepted definition of equivariance. It says that a function is equivariant if the equivariance error is below C, but equivariance should be reserved for exactly equivariant functions. The authors should reproduce the definition of Fei and Deng 2024 and name this C-weakly equivariance
- I found overall section 4.2 confusing:
    - I don't understand the justification for the encoder-decoder setup in 4.2
    - Equation 1 is simply training with data augmentation, that should be noted
    - The loss in equation 2 is an upper bound on the equation 2 loss (can the calculation be provided in the Appendix?). Minimizing 2 is not equivalent to minimizing 1, and will in general lead to a suboptimal minimum with respect to the original loss. The analysis that follows equation 2 does not take properly account for that.
    - If I understand correctly the analysis of equations 5 and 7 says that the strength of equivariance regularization is reduced when the input exhibits symmetry and that the model focuses on reducing prediction error. I don't think that's true. The bound says that the model will replace equivariance regularization with invariance regularization (which is just equivariant regularization with a different group action that preserves the symmetry) instead. So this does not solve the symmetry breaking problem.
- At the start of section 4.2 it is said that "establishing equivariance requires constraining all layers of the model". That's not true in general, a network can be equivariant without every layer being equivariant
- There's one thing I don't understand in the training process. For a symmetric shape, the pose is not determined by a unique rotation matrix. Yet training is done by regressing to a single rotation? Amongst the valid rotations, how will the network choose exactly the one used in the dataset?
- I think there are important issues regarding the experimental claims
    - I think the experiments are not representative of what the state of the art in pose estimation is. Saying that the method achieves "near SOTA" is not true at all. First, this is a toy task on a curated version of ModelNet. Second, the baselines considered are relatively simple and far from the state of the art.
    - Are non-equivariant models trained with data augmentation? It seems like something has gone wrong in training these models, the performance should not be so poor. I consider these results suspicious.

**Questions:**

- I think the decoupled vector linear layer is just a general linear layer, is that the case?

---

> ### Author Response · Authors · 2025-11-23
> **Response to Reviewer kASe - Part 1 Writing Rigor**
>
> **Dear Reviewer kASe**:
>
> We appreciate your comments! Your primary critiques center on three main aspects. The first concerns the mathematical theory (Section 4.2), specifically regarding the proof that optimizing the objective function reduces the model's equivariance error, and how the strength of equivariance regularization adaptively adjusts according to the symmetry of the input data. The second involves doubts regarding our experimental claims and results. The third points out that the writing is unclear or lacks rigor. We have rigorously revised the corresponding sections of the manuscript based on your feedback, and we are confident that these revisions will fully address your concerns. Below, please allow us to respond to your questions point-by-point in a new order.
>
>
> **Response to Comments on Writing Rigor**
>
> **W2:** Definition 4 is inconsistent with the accepted definition of equivariance. It says that a function is equivariant if the equivariance error is below C, but equivariance should be reserved for exactly equivariant functions. The authors should reproduce the definition of Fei and Deng 2024 and name this C-weakly equivariance.
>
> **Response to W2:** We appreciate the reviewer's careful observation.  This was indeed an oversight in our initial draft.  We have corrected the terminology to "C-weakly equivariance" in the revised manuscript to align with the definition provided by Fei and Deng (2024).
>
> **W4:** At the start of section 4.3 it is said that "establishing equivariance requires constraining all layers of the model". That's not true in general, a network can be equivariant without every layer being equivariant.
>
> **Response to W4:** We thank the reviewer for highlighting this issue. We acknowledge that our original phrasing was imprecise, as we omitted the specific context: **"within the VN framework."**Specifically for Vector Neuron Networks (VNNs), establishing equivariance indeed requires constraining all layers. Based on group theory, for a composite function $f \circ g$ to be equivariant, typically both $f$ and $g$ must be equivariant. In the VNN architecture, specific designs are implemented for every component, including VN-Linear, VN-Nonlinear (VN-ReLU, VN-LeakyReLU), and VN-Norm. If even a single layer is non-equivariant (e.g., using a standard ReLU or Linear layer), the chain of equivariance is broken. We have corrected the statement in the revised manuscript to read: "In the VN framework, establishing..." (Line 282).
>
>
>
> **Response to Comments on Decoupled Vector Linear Layer**
>
> **Q1:** I think the decoupled vector linear layer is just a general linear layer, is that the case?
>
> **Response to Q1:** The Decoupled Vector Linear Layer (DVLL) differs from a standard general linear layer in two key aspects:
>
> 1. Similar to the standard VN-Linear layer, the DVLL removes the bias term (transitioning from $y=wx+b$ to $y=wx$) to ensure that the individual linear operation remains equivariant.
> 2.  The DVLL operates independently on the three spatial components (channels) of the vector neurons.   It can be conceptualized as a stack of three bias-free linear layers applied separately to each spatial dimension.   Crucially, when the weights of these three linear layers are identical, the DVLL becomes mathematically equivalent to the standard VNN Linear layer.   This design allows us to effectively control the equivariance of Ada-VNN by adjusting the distance between the weights of these three channels.

---

> ### Author Response · Authors · 2025-11-23
> **Response to Reviewer kASe - Part 2 Theoretical Derivations and Proofs**
>
> **Response to Comments on Theoretical Derivations and Proofs**
>
> **W3:**  I found overall section 4.2 confusing:
>
> **Response to W3:** We appreciate your incisive comments! We acknowledge that there were indeed flaws in the mathematical theory (Section 4.2) of our initial submission. The main issues were: 1.  When the equivariance error serves as an upper bound for the objective function, optimizing the objective function does not necessarily imply a reduction in the equivariance error. 2.  It was not clearly explained how the strength of equivariance regularization varies when data exhibits different symmetries, preventing a direct conclusion regarding adaptive equivariance. We have extensively revised Section 4.2 in accordance with your feedback. We believe it now resolves your concerns. We will first briefly summarize the conclusions of the revised derivation, followed by a point-by-point response to your questions.
>
>
>
> **Revised Theoretical Derivation (Condensed Version of Section 4.2):** We aim to demonstrate that optimizing $\mathcal{L}$ forces a reduction in the model's equivariance error $\mathcal{E}(f)$ and enables adaptivity based on the data's intrinsic symmetry.
>
> **Preliminary Definitions**
>
> Equivariance Error ([1\*]): $\mathcal{E}(f) = \mathbb{E} _ {x,g}[\| f(\rho(g) x) - \rho(g) f(x) \|]$.
>
> We introduce the Symmetry Breaking Assumption ([2*]) as a replacement for the Haar measure:: The inherent symmetry breaking of the true function (the real physical process)  $f^\ast$ is bounded by $\epsilon$, i.e., $\sup _ {x,g}\| f^\ast(\rho(g) x) - \rho(g) f^\ast(x) \| = \epsilon$.
>
> Optimization Objective: $\mathcal{L} = \mathbb{E} _ {x,g}[\| f(\rho(g) x) - \rho(g) y \|]$.
>
> **Step 1: Decomposition of the Equivariance Error.**
> Using the triangle inequality of norms, we decompose $\mathcal{E}(f)$:
> $$
> \begin{aligned}
> \mathcal{E}(f) &= \mathbb{E} _ {x,g}[\| f(\rho(g) x) - \rho(g) f(x) \|] \\
> &= \mathbb{E} _ {x,g}[\| f(\rho(g) x) - \rho(g)f^\ast(x) + \rho(g)f^\ast(x) - \rho(g) f(x) \|] \\
> &\leq \mathcal{L} + \mathbb{E} _ {x,g}[\| \rho(g)f^\ast(x) - \rho(g)f(x) \|]
> \end{aligned}
> $$
>
> Since $\rho(g) \in \text{SO(3)}$ is a norm-preserving transformation, the second term simplifies to $\mathbb{E} _ {x}[\| f^\ast(x) - f(x) \|]$. Thus:
>
> $$
> \mathcal{E}(f) \leq \mathcal{L} + \underbrace{\mathbb{E} _ {x}[\| f^\ast(x) - f(x) \|]} _ {\text{Residual}}
> $$
>
> **Step 2: Relationship between residual term and loss function.**
> We expand $\mathcal{L}$ using the reverse triangle inequality:
> $$
> \mathcal{L} \geq \mathbb{E} _ {x,g} [ \| f(\rho(g) x) - f^\ast(\rho(g)x) \| ] - \underbrace{\mathbb{E} _ {x,g} [ \| f^\ast(\rho(g)x) - \rho(g)f^\ast(x) \| ]} _ {\leq\epsilon}
> $$
> For the first term, let $z = \rho(g)x$. We utilize the Volume Invariance of $\rho(g)$ ($|\det(\rho)|=1$) to preserve the integration measure ($dz=dx$). Combined with the standard assumption that the training distribution covers the group orbit (achieved via augmentation), we have the change-of-variables equality: $\mathbb{E} _ {x,g}[\| f(\rho(g)x) - f^\ast(\rho(g)x) \|] = \mathbb{E} _ {x}[\| f(x) - f^\ast(x) \|]$.
>
> Substituting this into the inequality yields: $\mathbb{E} _ {x}[\| f(x) - f^\ast(x) \|] \leq \mathcal{L} + \epsilon$
>
> **Step 3: Synthesis.**
> Substituting the result from Step 2 into Step 1, we obtain the bound $\mathcal{E}(f) \leq 2\mathcal{L} + \epsilon$.
> Considering the Ada-VNNs structure $f(x) = \Phi(x) + \Psi(\Phi(x))$, since the backbone $\Phi$ is strictly equivariant (satisfying $\Phi(\rho(g) x) = \rho(g) \Phi(x)$), the overall equivariance error is entirely determined by the residual part $\Psi$:
> $$
> \mathcal{E}(f) = \mathcal{E}(\Psi) \leq 2\mathcal{L} + \epsilon
> $$
>
> The proof is completed. This inequality demonstrates that by minimizing the proxy loss $\mathcal{L}$, the model's equivariance error is constrained within $2\mathcal{L} + \epsilon$. This reveals an adaptive mechanism:
>
> * When data has no symmetry breaking ($\text{Stab} _ {G}(y) \supseteq \text{Stab} _ {G}(x), \ \epsilon \rightarrow 0$): As $f$ is trained to convergence ($\mathcal{L} \to 0$), $\mathcal{E}(f) \to 0$, and $f$ tends to converge to a strictly equivariant model.
> * When data exhibits symmetry breaking ($\text{Stab} _ {G}(y) \nsupseteq \text{Stab} _ {G}(x),\ \epsilon>0$): As $f$ is trained to convergence ($\mathcal{L} \to 0$), $\mathcal{E}(f) \to \epsilon$, and $f$ tends to converge to an $\epsilon$-weakly equivariant model. This indicates that our optimization objective enables the model $f$ to achieve adaptive equivariance based on the data's symmetry properties.
>
> Notably, the derivation results align perfectly with our experimental findings (Sections 5.1 and 5.2). Due to space limitations, we recommend referring to the revised manuscript for further details on the derivation.
>
> **Below, we will provide a point-by-point response to your comments based on this revised theoretical derivation.**

---

> ### Author Response · Authors · 2025-11-23
> **Response to Reviewer kASe - Part 3 Theoretical Derivations and Proofs**
>
> **W3.1:** I don't understand the justification for the encoder-decoder setup in 4.2
>
> **Response to W3.1:** In the initial submission, we framed the mathematical modeling within the context of a rotation prediction task: the proposed model served as an encoder to extract features, followed by an MLP decoder to predict the rotation. In the revised version, we have removed the decoder $D$ from the modeling process, as the Lipschitz property allows this term to be eliminated without affecting our results. This simplification enables us to present the derivation process more clearly and concisely
>
> **W3.2:** Equation 1 is simply training with data augmentation, that should be noted.
>
> **Response to W3.2:** We appreciate you pointing this out. We have explicitly clarified in the revised manuscript that Equation 1 corresponds to training with data augmentation (Line 248).
>
> **W3.3:** The loss in equation 2 is an upper bound on the equation 2 loss (can the calculation be provided in the Appendix?). Minimizing 2 is not equivalent to minimizing 1, and will in general lead to a suboptimal minimum with respect to the original loss.
>
> **Response to W3.3:**
>
> We acknowledge that the proof provided in our initial submission contained errors. To rigorously demonstrate that minimizing the objective function minimizes the equivariance error, a tighter bound was required. As mentioned above, we have introduced a tighter constraint in the updated Section 4.2: $\mathcal{E}(f) \leq 2\mathcal{L} + \epsilon$. This inequality clearly and intuitively demonstrates that as the objective function is minimized ($\mathcal{L} \to 0$), the model's equivariance error is necessarily compressed to a value bounded by $\epsilon$ (where $\epsilon$ is not a variable, but a constant determined by the inherent symmetry of the data).
>
> **W3.4**: If I understand correctly the analysis of equations 5 and 7 says that the strength of equivariance regularization is reduced when the input exhibits symmetry and that the model focuses on reducing prediction error. I don't think that's true. The bound says that the model will replace equivariance regularization with invariance regularization (which is just equivariant regularization with a different group action that preserves the symmetry) instead. So this does not solve the symmetry breaking problem.
>
> **Response to W3.4:**  We are extremely grateful for this critical comment, as it was crucial in helping us realize that using a measure-theoretic approach to derive the model's adaptive equivariance was inappropriate. Consequently, in the revised version, we have replaced the measure-theoretic formulation with a  distance-based representation: $\sup_{x,g}\| f^\ast(\rho(g) x) - \rho(g) f^\ast(x) \| = \epsilon$. Here, $\epsilon$ quantifies the degree of inherent symmetry breaking in the true physical process, which depends on the data distribution; its value is positively correlated with the magnitude of symmetry breaking. Based on the inequality constraint $\mathcal{E}(f) \leq 2\mathcal{L} + \epsilon$, we draw the following conclusions:
>
> When data has no symmetry breaking ($\text{Stab} _ {G}(y) \supseteq \text{Stab} _ {G}(x), \ \epsilon \rightarrow 0$): As $f$ is trained to convergence ($\mathcal{L} \to 0$), $\mathcal{E}(f) \to 0$, and $f$ tends to converge to a strictly equivariant model.
>
> When data exhibits symmetry breaking ($\text{Stab} _ {G}(y) \nsupseteq \text{Stab} _ {G}(x),\ \epsilon>0$): As $f$ is trained to convergence ($\mathcal{L} \to 0$), $\mathcal{E}(f) \to \epsilon$, and $f$ tends to converge to an $\epsilon$-weakly equivariant model.
>
> This indicates that our optimization objective enables the model $f$ to achieve adaptive equivariance based on the data's symmetry properties, effectively allowing us to solve the symmetry breaking problem adaptively.

---

> ### Author Response · Authors · 2025-11-23
> **Response to Reviewer kASe - Part 4 Experimental Settings and Results**
>
> **Response to Comments on Experimental Settings and Results**
>
> **W6:**  I think there are important issues regarding the experimental claims
>
> **W6.1:** I think the experiments are not representative of what the state of the art in pose estimation is. Saying that the method achieves "near SOTA" is not true at all. First, this is a toy task on a curated version of ModelNet. Second, the baselines considered are relatively simple and far from the state of the art.
>
> **W6.2:** Are non-equivariant models trained with data augmentation? It seems like something has gone wrong in training these models, the performance should not be so poor. I consider these results suspicious.
>
> **Response to W6:** We believe there may be a misunderstanding regarding the positioning of our method and the critical problem we aim to solve. This has likely led to your doubts concerning our experimental design and results. Please allow us to clarify this aspect first, before addressing W6.1 and W6.2 individually.
>
> Our method is positioned as an **adaptive equivariant model for the SO(3) group, rather than a specialized model designed for the specific task of pose estimation.**
>
> The core problem addressed in this paper is that symmetries in real-world physical processes often exhibit symmetry breaking. Consequently, utilizing fully equivariant models or relying on fixed approximate equivariance is sub-optimal. We aim to employ an equivariant neural network with adaptive characteristics to better accommodate the varying degrees of symmetry breaking found in real physical processes.
>
> Our experiments focus on discussing two core results:
> 1.  Whether Ada-VNNs can achieve adaptive equivariance across different degrees of symmetry breaking (Sections 5.1 and 5.2 of the revised manuscript).
> 2.  Whether Ada-VNNs can demonstrate superior performance in the presence of symmetry breaking due to this adaptive mechanism (Section 5.3 of the revised manuscript).
>
> Therefore, we selected rotation prediction (pose estimation) as the validation task for the following reasons:
>
> 1. Different object categories exhibit varying degrees of symmetry breaking in SO(3) for rotation prediction tasks (higher self-symmetry implies a larger degree of symmetry breaking). This fits perfectly for verifying the adaptive characteristics of Ada-VNNs.
>
> 2. Equivariant neural networks are frequently used as backbones for rotation prediction because they provide intermediate features consistent with physical laws. Previous works on SO(3) equivariant networks (e.g., EPN, E2PN, VNN) also utilize rotation prediction as a comparative experiment. This follows the linear probing [7*] paradigm, comparing which representations are more beneficial for the task.
>
> 3. Rotation estimation is one of the most fundamental tasks in the 3D domain, widely used in SLAM, 3D reconstruction, scene understanding, and robotics. Thus, investigating whether Ada-VNN's representations are beneficial for this task holds significant practical value.
>
> **Below, we address W6.1 and W6.2**

---

> ### Author Response · Authors · 2025-11-23
> **Response to Reviewer kASe - Part 5 Experimental Settings and Results**
>
> **Response to W6.1**: First, our method is not a dedicated pose estimation system. Dedicated pose estimation methods typically require specific module designs, such as feature matching [3*], descriptors [4*], ICP optimization [5*], or coarse-to-fine strategies [6*]. However, works in the field of SO(3) equivariance generally do not utilize these modules. **The focus of SO(3) equivariance research is whether the model can produce representations conducive to pose estimation, rather than competing on leaderboard performance for pose estimation tasks.** Comparisons in this field often adopt a method similar to linear probing [7*]: the proposed model is used as a feature extractor, followed by a simple MLP or even a linear layer to regress rotation parameters directly from the features. This allows for a fair comparison of the quality of the model representations.
>
> Second, regarding the dataset, we utilized ShapeNet rather than ModelNet. We chose ShapeNet because it contains 55 different categories of point clouds. Different object categories exhibit varying degrees of symmetry breaking in SO(3) for rotation prediction tasks. We selected 26 categories covering different symmetries to verify the adaptive equivariant characteristics of Ada-VNNs. Therefore, for our method and other SO(3) equivariant neural networks, it is not necessary to select datasets designed for pose estimation tasks in difficult scenarios.
>
> Finally, regarding the "SOTA" claim: In our initial submission, the only mention of "SOTA" appeared in Section 5.2: *"Ada-VNT achieves near-SOTA results across symmetric categories while maintaining strong performance on asymmetric ones."* Our original intention was to express that, compared to other equivariant neural network, we achieved the best performance on categories with symmetry breaking while maintaining high performance on those without, **implying that Ada-VNT's representations are more beneficial for pose estimation tasks in the presence of symmetry breaking.** We did not mean that our method achieves SOTA in the general field of pose estimation. To avoid ambiguity, we have revised this sentence in the updated manuscript to: "Compared to other equivariant neural network, Ada-VNT achieves better results across symmetric categories, indicating that Ada-VNT provides representations more conducive to rotation estimation under symmetry breaking."
>
> **Response to W6.2**:First, regarding the experimental comparison, all methods were trained using a fully unified framework and identical data augmentation protocols, ensuring complete fairness. Our experimental framework aligns exactly with the settings established in EPN and E2PN.
>
> Second, we fully appreciate your skepticism; we were initially surprised by these results as well. However, after rigorous code reviews and comparative verification, we have confirmed the validity of the results. In this regard, we are grateful for your query, as it provides an opportunity to clarify this aspect. Furthermore, our code was open-sourced with the initial submission and supports the full reproduction of these experimental results.
>
> Finally, please allow us to briefly clarify the findings:
> It is, in fact, reasonable that PointNet++ and PCT yield near-collapse results in this specific setting.  **As previously mentioned, existing methods that successfully utilize PointNet++ or PCT for pose estimation typically incorporate additional specialized modules, such as feature matching [3\*], descriptors [4\*], ICP optimization [5\*], or coarse-to-fine strategies [6\*].**  These modules are critical for pose estimation because neural network-based approaches inherently evolved from geometric optimization frameworks (see [8*]), often replacing explicit matching and optimization steps with data-driven components. Consequently, directly regressing rotation parameters via a simple MLP from the representations of fully non-equivariant models like PointNet++ and PCT is inherently difficult, as these models lack any inductive bias regarding symmetry.  This challenge is exacerbated by the fact that our experiment targets category-level rotation estimation rather than instance-level alignment for a single specific point cloud.
>
> We thank you for your feedback!  In the revised manuscript, we have added a new section (**Sec.  5.4, "Why Ada-VNNs Work?"** ) to further clarify why our results outperform both fully equivariant and fully non-equivariant neural networks in this context.

---

> ### Author Response · Authors · 2025-11-23
> **Response to Reviewer kASe - Part 6 Experimental Settings and Results**
>
> **W5:** There's one thing I don't understand in the training process. For a symmetric shape, the pose is not determined by a unique rotation matrix. Yet training is done by regressing to a single rotation? Amongst the valid rotations, how will the network choose exactly the one used in the dataset?
>
> **Response to W5:** This question is critical, as it touches upon the core mechanism underlying the effectiveness of our method.
>
> First, we can understand this issue by distinguishing between shape similarity (e.g., Chamfer Distance) and angular similarity (e.g., Angular Error). For a given shape, there may exist multiple rotation matrices $[R_{1}, R_{2}, \dots, R_{n}]$ that achieve sufficiently small Chamfer Distance errors (e.g., for an object approximating a rectangular cuboid). However, only one rotation matrix $R_{1}$ corresponds to the minimum angular error.
>
> In the extreme case of a perfectly symmetric object, such as a perfect sphere, no deterministic function can recover the unique ground truth $R_{1}$ from the input. However, in reality, object point clouds typically exhibit **approximate symmetry rather than perfect symmetry**. They contain high-frequency geometric details that break perfect symmetry, such as the pull-tab on a soda can, the side buttons and rear camera on an iPhone, or even surface textures and random noise from point cloud sampling. In these cases, the object is approximately symmetric with respect to a symmetry subgroup $G_s$ (i.e., $\forall g_s \in G_s, \rho(g_s)x \approx x$). Defining the difference as $\delta = \|\rho(g_s)x - x\|$, the closer the object is to perfect symmetry, the smaller $\delta$ becomes.  Hence, the task requires us to regress to a unique rotational solution. This can be explained using the analogy of a loaded die: a die is geometrically symmetric (cubic), but if a small amount of lead is injected inside, this minute mass offset determines its unique stable state when it comes to rest. Many physical processes follow this phenomenon.
>
> You might wonder: since the symmetry is not perfect, why does our method handle this problem better than strictly equivariant networks? We explain this further below.
>
> The inequality $\|\rho(g_s)f(x) - f(x)\| \leq k\delta$ implies that when the input is very close to symmetric ($\delta \to 0$), the output feature $f(x)$ is forced to collapse toward the Fixed Point Set of the symmetry group $G_s$.
>
> If the symmetry subgroup $G_s$ represents continuous rotational symmetry about a specific axis, the only vectors satisfying the approximate invariance condition (i.e., the Fixed Point Set) are those aligned with this rotation axis. Consequently, the components of $f(x)$ orthogonal to the rotation axis-which carry the critical geometric information for distinguishing orientation-are forced to approach zero magnitude. This **feature collapse** erases the necessary signal to resolve directional ambiguity, causing the regression layer to fail in predicting the rotation."
>
> In contrast, Ada-VNNs are not bound by the strict equality constraint mentioned above. This means that even if $\delta$ is small, Ada-VNNs are allowed to produce features where $\rho(g_s)f(x) \neq f(\rho(g_s)x)$. This flexibility enables the model to amplify and exploit those tiny "asymmetries" to determine the unique state. To enhance clarity, **we have added Section 5.4 in the revised manuscript to discuss this mechanism and included visualizations of feature collapse in Figure 5.**

---

> ### Author Response · Authors · 2025-11-23
> **Response to Reviewer kASe - Part 7   Core Differences Between Ada-VNNs and Wang et al. (2022)**
>
> **Response to Comments on Core Differences Between Ada-VNNs and Wang et al. (2022)**
>
> **W1:** I think the contributions of this paper are relatively incremental with respect to the existing works on relaxing equivariance works, especially Wang et al. 2022. The method proposed in this paper is essentially an adaptation of Wang et al 2022 to the Vector Neuron architecture.
>
> **Response to W1**:  We thank the reviewer for raising this critical point. We hold the pioneering work of Wang et al. (2022) in high regard and acknowledge it as a source of inspiration for our research. However, we respectfully point out that **Ada-VNNs is by no means a simple adaptation of their method to the Vector Neuron architecture.** The two approaches differ fundamentally in the following five aspects:
>
> 1. **Distinct Mechanisms for Symmetry Breaking.** Wang et al. (2022) achieve this by relaxing the weight-sharing constraint of convolution kernels  (e.g., making kernel weights dependent on group element pairs $(g, h)$ rather than just $g^{-1}h$). In contrast, Ada-VNNs introduce a residual architecture, where a symmetry-breaking lightweight branch $\Psi$ is appended to a strictly equivariant backbone $\Phi$ ($f(x) = \Phi(x) + \Psi(\Phi(x))$). Furthermore, we design the Decoupled Vector MLP (DV-MLP), which employs three independent weight matrices ($W^1, W^2, W^3$). Symmetry breaking is controlled by the divergence among these matrices, a mechanism specific to  Ada-VNNs.
> 2. **Distinct Mechanisms for Adaptivity (Implicit vs. Explicit).** Wang et al. (2022) rely on explicit regularization, requiring an additional penalty term in the loss function (e.g., $||w_i(h) - w_i(g)||$) and manual tuning of hyper-parameters. Ada-VNNs utilize implicit regularization: we theoretically prove that no additional equivariance penalty is needed. By simply minimizing the task prediction loss $\mathcal{L}$, the model's equivariance error $\mathcal{E}(f)$ is automatically bounded by the data's intrinsic symmetry breaking $\epsilon$.
> 3. **Different Theoretical Focus.** Wang et al. (2022) primarily prove that strictly equivariant models cannot approximate approximately equivariant ground truths. Building upon this, Ada-VNNs go further to derive an adaptive mechanism directly from the objective function. Additionally, we provide a theoretical analysis of  feature collapse, explaining why strictly equivariant models fail in symmetry-breaking tasks
> 4. **Different Base Architectures and Groups.** Wang et al. (2022) are based on Convolutional Neural Networks (Group/Steerable CNNs), focusing primarily on 2D planar symmetry groups (e.g., $SO(2)$). Ada-VNNs are built upon the Vector Neuron framework, specifically addressing the 3D rotation group $SO(3)$.
> 5. **Different Application Domains and Data Objects.** Wang et al. (2022) primarily process 2D vector fields or grid-based data (e.g., smoke simulations, turbulent velocity fields, and ocean currents) , whereas Ada-VNNs mainly handle 3D data (e.g., point clouds).
>
> Finally, we appreciate your recognition of our work's contribution. Scientific progress is often built upon the cumulative efforts of the community, expanding on previous foundations. We believe our work represents a meaningful step forward in this direction and hope to contribute to the continued advancement of the geometric deep learning community.
>
> **Closing Remarks**
>
> Once again, we extend our sincere gratitude for your constructive criticism, which has been instrumental in enhancing the quality of our work.  We believe the revised manuscript now offers a mathematically rigorous and empirically consistent explanation for the adaptive equivariance of Ada-VNNs.  We hope these comprehensive revisions and detailed responses satisfactorily address your concerns. Finally, we deeply appreciate the time you dedicated to reviewing our manuscript and your contribution to the open AI community.  As fellow researchers, we also wish you the best of luck with your own rebuttals during this season.  We look forward to any further discussion during the interaction period and remain fully committed to addressing any remaining questions.

---

> ### Author Response · Authors · 2025-11-23
> **Response to Reviewer kASe - Part 8 References**
>
> **References**
>
> [1\*] Fei, Jiajun, and Zhidong Deng. "Rotation invariance and equivariance in 3D deep learning: a survey." Artificial Intelligence Review 57.7 (2024): 168.
>
> [2\*] Wang, Rui, Robin Walters, and Rose Yu. "Approximately equivariant networks for imperfectly symmetric dynamics." International Conference on Machine Learning. PMLR, 2022.
>
> [3*] Li Y, Harada T. Lepard: Learning partial point cloud matching in rigid and deformable scenes[C]//Proceedings of the IEEE/CVF conference on computer vision and pattern recognition. 2022: 5554-5564.
>
> [4*] K. Fischer, M. Simon, F. Ölsner, S. Milz, H. -M. Groß and P. Mäder, "StickyPillars: Robust and Efficient Feature Matching on Point Clouds using Graph Neural Networks," *2021 IEEE/CVF Conference on Computer Vision and Pattern Recognition (CVPR)*, Nashville, TN, USA, 2021, pp. 313-323, doi: 10.1109/CVPR46437.2021.00038.
>
> [5*] He, Leping, et al. "Gfoicp: Geometric feature optimized iterative closest point for 3-d point cloud registration." *IEEE Transactions on Geoscience and Remote Sensing* 61 (2023): 1-17.
>
> [6*] Qin, Zheng, et al. "Geometric transformer for fast and robust point cloud registration." *Proceedings of the IEEE/CVF conference on computer vision and pattern recognition*. 2022
>
> [7*] He, Kaiming, et al. "Masked autoencoders are scalable vision learners." *Proceedings of the IEEE/CVF conference on computer vision and pattern recognition*. 2022.
>
> [8*] Yang, Jiaqi, et al. "3d registration in 30 years: A survey." *arXiv preprint arXiv:2412.13735* (2024).

---

### Note · Authors · 2026-01-30

I have read and agree with the venue's withdrawal policy on behalf of myself and my co-authors.

---

### Meta-Review · Area_Chair_hTBD · 2026-01-11

**Summary:**

There was significant discussion on this paper, particularly with the reviewers who tended to be more critical. During the review, some technical issues with the proof were found. This was discussed, and possibly addressed, during the rebuttal period. However, I agree that another round of reviewing is needed to determine whether this paper should be accepted. The recommendation for rejection is not _because_ the authors engaged with the rebuttal process. It is indeed in the spirit of ICLR to encourage authors to improve their work. Hopefully the paper is better, and I hope the authors of the paper found this a valuable process. However, the review process clearly states that there are limits to what can be asked of the reviewing team, and that sometimes means that papers need to be resubmitted.

**Reviewer Concerns:**

The most important concern that was addressed was a flaw in the theory section of the paper. This resulted in a significant new part of the paper. This was not a case where the _existing_ paper was correct, and clarified in the review process, but instead where the reviewers highlighted a legitimate concern.

I thank the reviewers for engaging so helpfully with the authors, to help them to improve the paper. I hope the authors found this process helpful, and that their paper has improved for the next review cycle.

**Reviewer Scores:**

The reviewers with positive scores did not engage much with the discussion, but the reviewers with negative scores significantly did (both with the authors, and afterwards). As such, it is unlikely that the negative scores would have increased to the degree needed for acceptance.

---

### Decision · Program_Chairs · 2026-01-26

Reject